# Single-cell allele-specific expression analysis reveals dynamic and cell-type-specific regulatory effects

Guanghao Qi [1,2], Benjamin J. Strober [3], Joshua M. Popp[1], Rebecca Keener [1], Hongkai Ji [4] & Alexis Battle [1,5,6] ✉

Differential allele-specific expression (ASE) is a powerful tool to study context-specific cis-regulation of gene expression. Such effects can reflect the interaction between genetic or epigenetic factors and a measured context or condition. Single-cell RNA sequencing (scRNA-seq) allows the measurement of ASE at individual-cell resolution, but there is a lack of statistical methods to analyze such data. We present Differential Allelic Expression using Single-Cell data (DAESC), a powerful method for differential ASE analysis using scRNA-seq from multiple individuals, with statistical behavior confirmed through simulation. DAESC accounts for non-independence between cells from the same individual and incorporates implicit haplotype phasing. Application to data from 105 induced pluripotent stem cell (iPSC) lines identifies 657 genes dynamically regulated during endoderm differentiation, with enrichment for changes in chromatin state. Application to a type-2 diabetes dataset identifies several differentially regulated genes between patients and controls in pancreatic endocrine cells. DAESC is a powerful method for single-cell ASE analysis and can uncover novel insights on gene regulation.

Allele-specific expression (ASE) measures the expression of one allele of a gene relative to the other in a diploid individual. ASE is a powerful tool to study allelic imbalance caused by cis-regulatory genetic variation[1–3] and epigenetic alterations such as imprinting[4]. In particular, heterozygous expression quantitative trait loci (eQTLs) variants in or near a gene can cause two alleles to be expressed at different levels[1,2]. Compared to standard eQTL testing, ASE is less susceptible to some confounders, including environmental and technical conditions. In addition, comparison of ASE across conditions (differential ASE) can reveal context-specific cis-regulatory effects. Previous ASE studies found that regulatory effects can vary by smoking status[5], blood pressure medication usage[5], and stages of CD4 + T-cell activation[6], among others.

ASE has been extensively explored using bulk RNA sequencing, but this cannot capture heterogeneity across cell types within a tissue.

Recently, single-cell RNA sequencing (scRNA-seq) has enabled the quantification of ASE at the resolution of individual cells[7–10] (Fig. 1a), often across multiple individuals. In this paper, we focus on identifying genes that show differential ASE across conditions. Related methods are only beginning to emerge, and previous approaches are currently applicable to a limited set of scenarios due to assumptions of the models[11,12]. scDALI[11] uses a beta-binomial mixed-effects model to detect differential allelic imbalance across discrete cell types or continuous cell states. Another method, airpart[12], partitions the data into groups of genes and cells with similar patterns of allelic imbalance. Airpart also has a function for differential ASE testing based on a hierarchical Bayesian model[12].

However, scDALI and airpart do not account for some experimental designs that include scRNA-seq data from multiple individuals.

[1]Department of Biomedical Engineering, Johns Hopkins University, Baltimore, MD 21218, USA. [2]Department of Biostatistics, University of Washington, Seattle, WA 98195, USA. [3]Department of Epidemiology, Harvard T.H. Chan School of Public Health, Boston, MA 02115, USA. [4]Department of Biostatistics, Johns Hopkins Bloomberg School of Public Health, Baltimore, MD 21205, USA. [5]Department of Computer Science, Johns Hopkins University, Baltimore, MD 21218, USA. [6]Department of Genetic Medicine, Johns Hopkins University, Baltimore, MD 21205, USA. ✉e-mail: ajbattle@jhu.edu

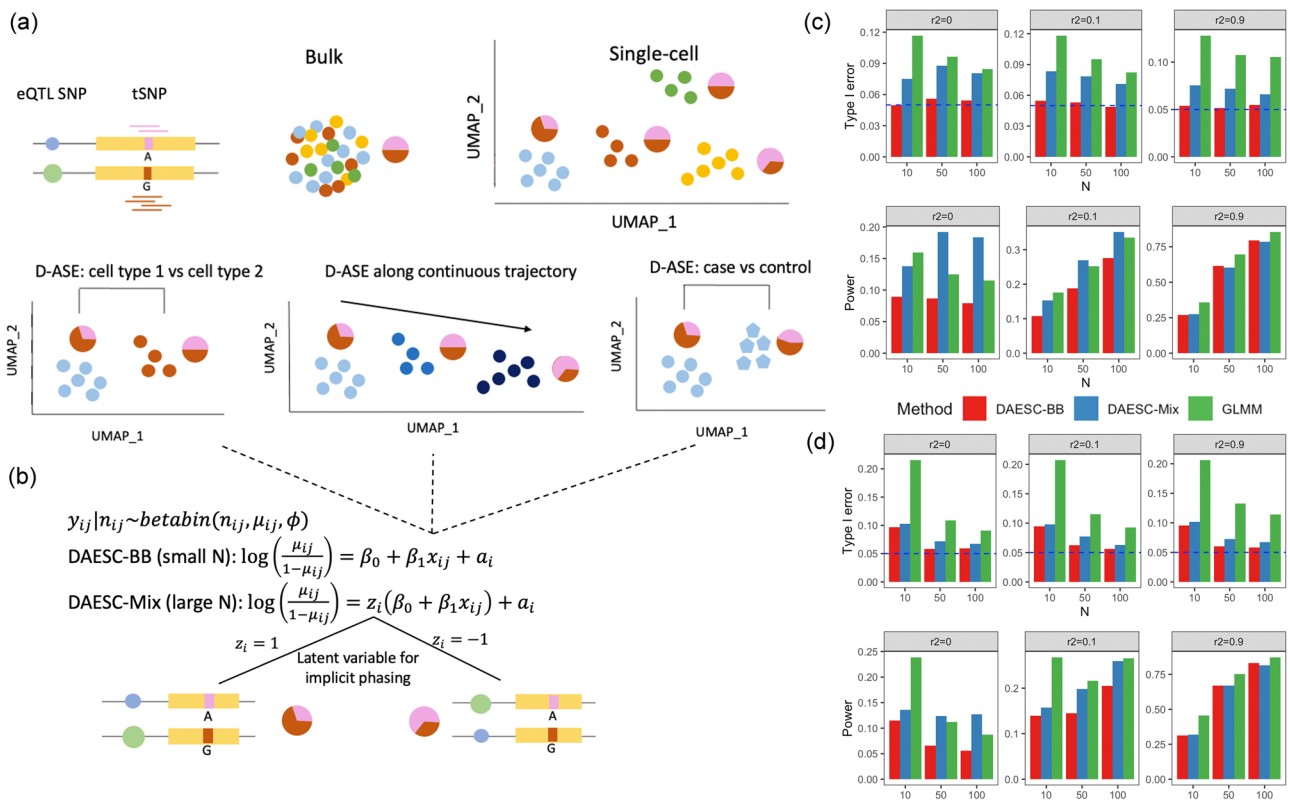

**Fig. 1 | Schematic of DAESC and simulation studies. a** Schematic of allele-specific expression (ASE) measured in bulk tissue and single cells, and three types of differential ASE analysis. Pie charts represent the relative expression of two alleles of the transcribed SNP (tSNP). **b** DAESC models. DAESC accounts for sample repeat structure (multiple cells per sample) using random effects $a_i$ and implicit haplotype phasing using latent variables $z_i$. **c, d** Type I error and power observed in simulation studies for differential ASE, **c** along a continuous cell state and **d** binary case–control disease status. Type I error and power are computed under nominal significance threshold $P < 0.05$ (no multiple comparison adjustment). Likelihood ratio test is used for DAESC-BB and DAESC-Mix and $z$-test is used for GLMM. All tests are two-sided. Allele-specific read counts are simulated from beta-binomial mixture model assuming only one eQTL drives ASE at a tSNP. The linkage disequilibrium between the eQTL and the tSNP is varied to $r^2 = 0, 0.1, 0.9$, and the sample size (number of individuals) is varied to $N = 10, 50, 100$. Source data are provided as a Source Data file.

One major challenge arising from multiple individuals is how to align read counts consistently across individuals. In the eQTL setting, for example, a noncoding eQTL variant that drives ASE is not observed. Specifically, its expression-increasing allele can be on the haplotype of either the alternative or the reference allele of the exonic SNP where ASE is assessed (transcribed SNP or tSNP, Fig. 1b)[5,13,14]. As a result, different individuals may have opposite allelic imbalance at the tSNP, but actually representing a consistent genetic regulatory effect. We refer to this phenomenon as "haplotype switching" in the rest of the paper. If not addressed, the allelic imbalance observed across individuals will cancel out, leading to diminished signal. This issue also exists for ASE caused by epigenetic factors. Previous cross-individual ASE methods for bulk RNA-seq used a majority voting approach, which treats the lower allelic read count as the alternative allele read count[5,14]. However, this approach is not applicable to single-cell ASE due to low total read count per cell. The scDALI paper avoided this issue with an extra step in the preprocessing, by using phased genotype data and pre-identified eQTLs to align read counts[11]. This approach is not applicable to general differential ASE settings where genotypes are not available or where no significant eQTL has been identified for the gene. A second challenge arising from scRNA-seq data of multiple individuals is the sample repeat structure caused by having multiple cells per individual. This can cause false positives if all cells are treated as independent[11]. scDALI and airpart can account for this structure by adjusting donor IDs as fixed-effects covariates[11,12]. However, this approach is not applicable to comparing ASE between groups of individuals, e.g., disease cases vs controls, since donor IDs are collinear

with the binary variable of disease status, and treating them as fixed effects could lead to misleading results.

We present Differential Allelic Expression using Single-Cell data (DAESC), a statistical framework for identifying genes with differential ASE using scRNA-seq data of multiple individuals applicable to a wide range of study designs. DAESC accounts for haplotype switching using latent variables and handles sample repeat structure of single-cell data using random effects. Simulation studies show the method has robust type I error and high power for differential ASE testing. Applied to single-cell ASE data of 105 individuals[10], DAESC identifies hundreds of genes with dynamic ASE during endoderm differentiation. Follow-up analyses using the Roadmap Epigenomic data[15] show that dynamic ASE is linked to changes in chromatin state. A second application to a small dataset[8] without genotype information identifies three genes with differential ASE in pancreatic endocrine cells between type 2 diabetes (T2D) patients and controls.

## Results

### Overview of DAESC

DAESC is based on a beta-binomial regression model and can be used for differential ASE against any independent variable $x_{ij}$, such as cell type, continuous developmental trajectories, genotype (eQTLs), or disease status (Fig. 1a). DAESC includes two versions (DAESC-BB and DAESC-Mix) to be used under different scenarios (Fig. 1b). The baseline model DAESC-BB is a beta-binomial model with individual-specific random effects ($a_i$) that account for the sample repeat structure ("Methods") arising from multiple cells measured per individual

inherent to single-cell data. DAESC-BB can be used generally for differential ASE regardless of sample size (number of individuals, $N$). When sample size is reasonably large (e.g., $N \geq 20$), we introduce a full model DAESC-Mix that accounts for both sample repeat structure and implicit haplotype phasing ("Methods"). For example, when ASE measured at a heterozygous tSNP is driven by an eQTL, the expression-increasing allele of the eQTL variant could be on either haplotype. We account for this possibility using latent variables $z_i$'s that conduct implicit phasing, which lead to a mixture model (Fig. 1b). Specifically, if the expression-increasing eQTL allele is on the haplotype of the alternative allele of the tSNP, the model reflects this with $z_i = 1$; if the eQTL allele is on the haplotype of the reference allele of the tSNP, the model reflects this with $z_i = -1$. Although it is possible that the true model may have more mixture components, especially when the gene has multiple eQTLs, we use the two-component mixture model to prevent overfitting and increase computational speed. For both DAESC-BB and DAESC-Mix, parameter estimation is conducted using a variational EM algorithm (see "Methods" and Supplementary Notes for details). Hypothesis testing for differential ASE ($H_0 : \beta_1 = 0$) is conducted using a likelihood ratio test.

## Simulation studies

We first conduct simulations from a beta-binomial mixture model assuming only one eQTL drives ASE at the tSNP. In the first scenario, we test differential ASE along a continuous variable representing cell state (e.g., differentiation stage), we observe that DAESC-BB has well-controlled type I error across scenarios (Fig. 1c). DAESC-Mix has slight type I error inflation (averaged 8.5% across scenarios) but less than a standard generalized linear mixed model (GLMM, averaged 10% across scenarios). If provided with enough computational resources, the users can choose to conduct permutation tests to further correct the type I error of DAESC-Mix. See Methods for formulation of the GLMM. When there is no LD between the eQTL and tSNP ($r^2 = 0$), we observe a substantial power gain by using DAESC-Mix compared to DAESC-BB and the GLMM. The gain is more pronounced when the sample size is large ($N = 50$ or $100$). This is likely due to the ability of DAESC-Mix to conduct implicit haplotype phasing, which was shown to be effective overall (Fisher's exact test $P$ value < 0.05 in 36.5% genes tested, Supplementary Fig. 1). When $r^2 = 0.1$, DAESC-Mix has similar power to the GLMM, and both are slightly more powerful than DAESC-BB. When the LD between the eQTL and tSNP is strong ($r^2 = 0.9$), we observe only minimal power difference across the three methods. Using eQTL data for whole blood from the GTEx Consortium[16] as an example, we show that LD $r^2 < 0.1$ for most eQTL-tSNP pairs (Supplementary Fig. 2), indicating that for most genes DAESC-Mix is likely to lead to improved power. For differential ASE with respect to binary case–control disease status, we observe mostly similar patterns as those in the previous simulation with continuous cell state (Fig. 1d). A notable distinction is that all methods have more inflated type I error (-10%) when $N \leq 10$, and the GLMM have higher type I error inflation across scenarios.

In addition to GLMM, we compare DAESC with other methods, including beta-binomial regression implemented by apeglm[17] (also used in airpart[12]), apeglm with donor IDs adjusted as covariates (apeglm-adj), EAGLE[5], and EAGLE applied to pseudobulk data (EAGLE-PB). See Methods for details. We observe inflated type I error for apeglm and EAGLE due to failure to account for the sample repeat structure (Supplementary Fig. 3). Apeglm-adj used fixed effects to account for sample repeat structure and have nearly identical performance as DAESC-BB for continuous cell states (Supplementary Fig. 3). However, it cannot be applied to case–control comparisons since the case–control variable is colinear with the one-hot encoding of donor IDs. EAGLE-PB, the pseudobulk-based method for case–control comparisons, is less powerful than DAESC-BB especially when $r^2 = 0.1$ and $0.9$ (Supplementary Fig. 3). This shows the advantage of directly analyzing single-cell data over pseudobulk aggregation. EAGLE-PB assumes independent samples and is not applicable to the continuous-cell-state simulations shown in Fig. 1c and Supplementary Fig. 3a. The precision–recall curves show that DAESC-Mix dominates the other methods when $r^2 = 0$ and $N \geq 50$ with varying significance thresholds (Supplementary Fig. 4), especially in the simulations for continuous cell states. In addition, the curves for the GLMM tend to dip near low recall value (Supplementary Fig. 4), i.e., when the significant threshold is stringent. This indicates potential issues with $P$ value calibration. Nevertheless, GLMM appears to be the most comparable to DAESC-BB considering type I error and power, and its applicability to both continuous cell state and case–control comparisons. We use GLMM as the main comparison for the rest of the simulation studies.

Since eQTL studies have found that allelic heterogeneity is widespread[18–21], we also investigate the performance of the methods when there are multiple eQTLs driving ASE. Due to the large number of scenarios for levels of LD across multiple eQTLs and the tSNP, we limit our investigation to the scenario where no LD exists between the eQTLs or between the eQTLs and the tSNP. Similar to the previous scenario, DAESC-BB controls type I error under varying numbers of eQTLs; DAESC-Mix has slightly inflated type I error in some settings, but is less inflated than the GLMM (Fig. 2a). This shows that although having multiple eQTLs introduces extra mixture components into the true model ("Methods"), it has minimal impact on the type I error control. Furthermore, we observe a substantial power gain by DAESC-Mix compared to DAESC-BB or the GLMM (Fig. 2a), which is more pronounced than when only one eQTL drives ASE (Fig. 1). This gain exists not only under a large sample size but also under small sample size ($N = 10$), although with a smaller margin. In addition, power increases steadily for DAESC-Mix with increasing number of eQTLs, showing a larger advantage over DAESC-BB and the GLMM under allelic heterogeneity (Fig. 2a). Precision–recall curves show that DAESC-Mix consistently outperforms the other two methods across different significance thresholds, with DAESC-BB ranking second (Fig. 2b). When testing differential ASE for binary case–control disease status, DAESC-Mix remains most powerful when there are multiple eQTLs per tSNP (Supplementary Fig. 5). In fact, DAESC-BB, the GLMM, and EAGLE-PB, which do not conduct implicit phasing, do not appear to have any power to detect differential ASE. In contrast to differential ASE along continuous cell state (Fig. 2), the power of DAESC-Mix changes minimally with the number of eQTLs (Supplementary Fig. 5).

Next, we investigate the performance of DAESC under varying data quality, which is reflected by overdispersion parameter ($\phi$) and sequencing depth. DAESC-BB and DAESC-Mix outperforms GLMM across varying levels of overdispersion (Supplementary Fig. 6). Although all methods have lower power under strong overdispersion (large $\phi$, low data quality), the advantage of DAESC of GLMM is also more pronounced (Supplementary Fig. 6). We also observe that though DAESC-Mix is developed for large $N$ (e.g., $N > 20$), it can also deliver strong performance under small $N$ when the overdispersion is low (e.g., $N = 6$ and $\phi = 0.5$, Supplementary Fig. 6), which is the case for many mouse datasets with low variance. In addition, we observe similar relative performance for DAESC-BB, DAESC-Mix, and GLMM under 50%, 20%, and 10% sequencing depth of other scenarios, though all methods have lower power (Supplementary Fig. 7).

To evaluate the sensitivity of DAESC to model misspecification, we conduct another simulation study using binomial GLMM instead of beta-binomial (see "Simulation studies"). Theoretically, this scenario should give more advantage to the GLMM method. However, DAESC-BB and GLMM have nearly identical performance (Supplementary Fig. 8). DAESC-Mix still leads to substantial power gain when there is low LD between the eQTL and the tSNP (Supplementary Fig. 8). This shows that DAESC has robust performance even when the beta-binomial assumption is violated. We observe that though DAESC is

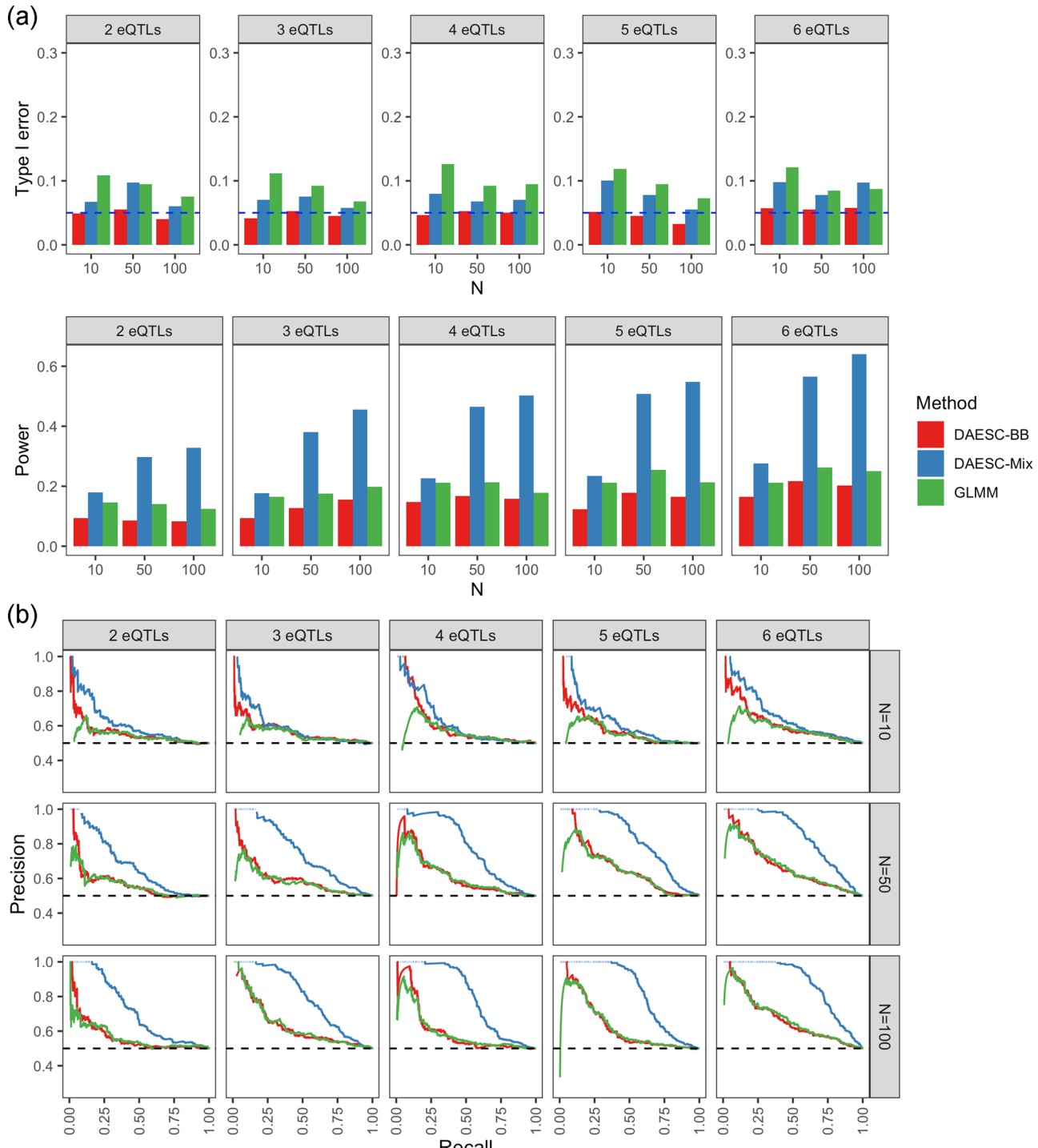

**Fig. 2 | Simulation studies with multiple eQTL SNPs per gene. a** Type I error and power and **b** precision–recall curves for differential ASE detection along a continuous variable observed in simulations. Allele-specific read counts are simulated from beta-binomial mixture model assuming multiple eQTLs drives ASE of a transcribed SNP (tSNP). We assume no linkage disequilibrium among the eQTLs and between the eQTLs and the tSNP. The sample size (number of individuals) is varied $N = 10, 50, 100$. Source data are provided as a Source Data file.

computationally intensive due to its EM iterations, it can be easily handled by a modern computing cluster (see "Methods" for details and Supplementary Fig. 9 for results). For example, when analyzing a dataset of 200 individuals and on average 400 cells per individual (> 2.5 times the size of the endoderm differentiation dataset[10] in our application), DAESC-BB requires 3.3 h to analyze 100 genes and DAESC-Mix requires 8.6 h (Supplementary Fig. 9).

**Dynamic ASE during endoderm differentiation**

We apply DAESC-BB, DAESC-Mix, and the GLMM to single-cell ASE data for 30,474 cells from 105 individuals collected by Cuomo et al.[10]. In their experiment, induced pluripotent stem cells (iPSCs) underwent differentiation for three days into mesendoderm and definitive endoderm cells (Fig. 3a). To study dynamic regulatory effects along the differentiation trajectory, we conduct differential ASE analysis along

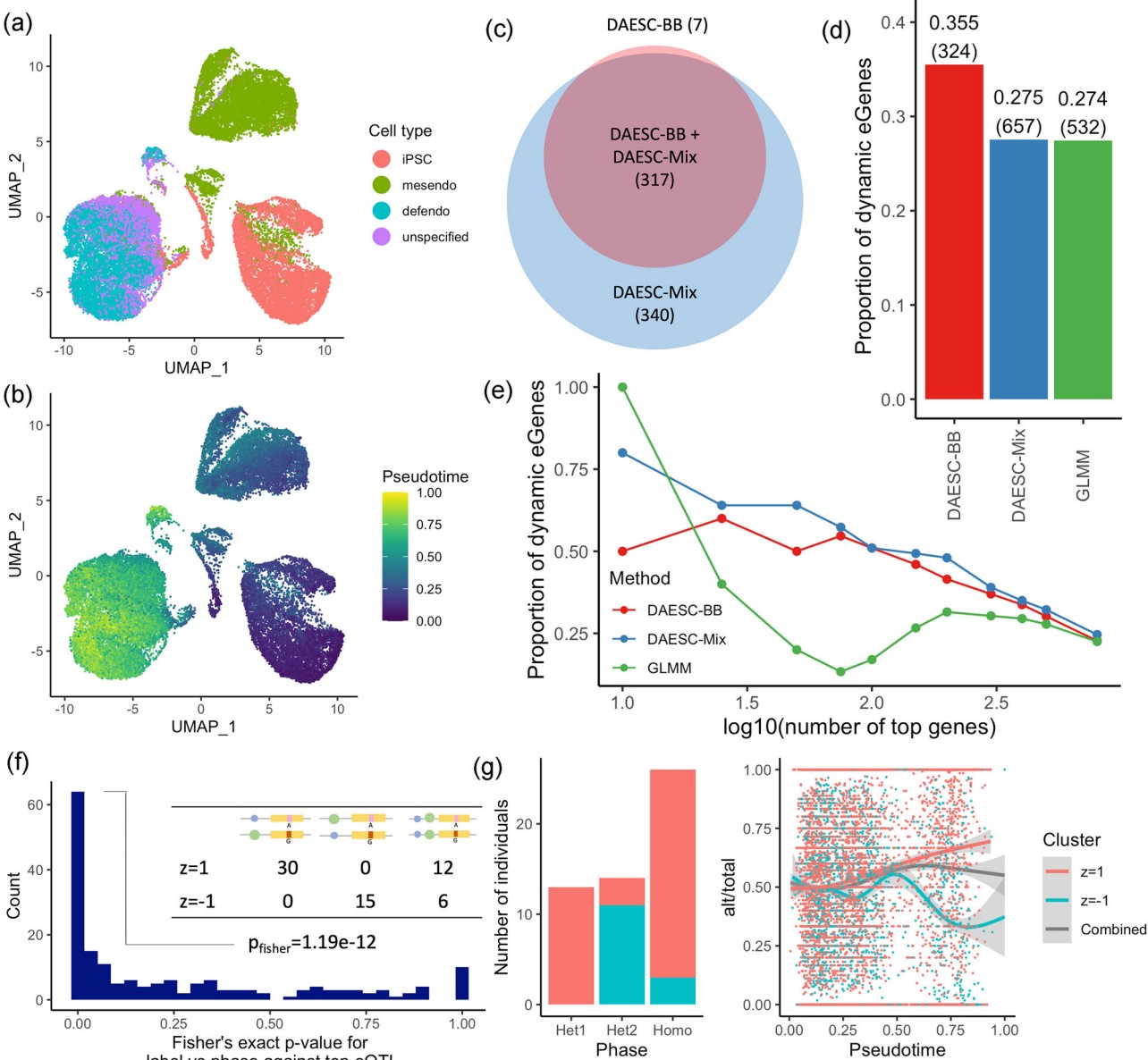

**Fig. 3 | Dynamic ASE during endoderm differentiation.** UMAP plot colored by **a** cell type and **b** pseudotime. Cell types include induced pluripotent stem cells (iPSCs), mesendoderm cells (mesendo) and definitive endoderm cells (defendo). **c** Venn diagram for the number of dynamic ASE (D-ASE) genes identified by DAESC-BB and DAESC-Mix. **d** Proportion of D-ASE genes identified by three methods that were also dynamic eGenes reported by Cuomo et al. (validation criterion). The number of D-ASE genes identified by each method are annotated in the parentheses. **e** Proportion of dynamic eGenes reported by Cuomo et al. among varying number of top D-ASE genes identified by three methods. **f** Two-sided Fisher's exact test $P$ values testing whether DAESC-Mix cluster labels capture haplotype

pseudotime ($x_{ij}$), which was estimated and provided by the original study (Fig. 3b).

DAESC-BB identifies 324 dynamic ASE (D-ASE) genes that vary along pseudotime and DAESC-Mix identifies 657 D-ASE genes (FDR < 0.05, Fig. 3c and Supplementary Data 1). Nearly all genes identified by DAESC-BB are also identified by DAESC-Mix (Fig. 3c). Since D-ASE can be driven by dynamic cis-regulatory effects, we use the overlap between our D-ASE genes and dynamic eQTL genes reported by Cuomo et al.[10] as a validation criterion. Among the genes identified by DAESC-BB, 35.5% were reported by Cuomo et al., while among those identified by DAESC-Mix 27.5% were reported (Fig. 3d).

information between the top tSNP and top eQTL reported by Cuomo et al. Schematics of three haplotype combinations are used as column names of the example 2 × 3 table (from left to right: het1, het2, homo). Green and blue circles are the reference (ref) and alternative (alt) alleles of the eQTL, respectively; red and pink rectangles are the alt and ref for the tSNP, respectively. **g** An example (*NMU* gene) of mixture clusters capturing haplotype information. Alt: alternative allele read count; total: total allele-specific read count. Trend curves are generated using ggplot2::geom_smooth() and shadings represents 95% confidence bands. Source data are provided as a Source Data file.

The GLMM identifies 19% fewer genes than DAESC-Mix (532 vs 657) and has a similar validation rate (Fig. 3d). Comparing the same number of top genes (by smallest $P$ values) selected by each method, DAESC-Mix shows a higher validation rate than DAESC-BB or the GLMM across varying number of top genes (Fig. 3e). The curve for GLMM dips sharply in the middle indicating $P$ value mis-calibration. scDALI finds 274 genes at FDR < 0.05, 77% of which are also found by DAESC-BB (Supplementary Fig. 10). In addition, dynamic ASE genes discovered using DAESC-Mix display total expression trends along pseudotime similar to those of previously discovered dynamic eQTL genes (Supplementary Fig. 11). This shows that DAESC-Mix

offers an increase in power without biasing discovery toward particular trends in expression or technical factors influencing total expression levels.

We conduct two sensitivity analyses to evaluate the effects of analysis choices on the results. First, in the main analysis, we remove SNPs with monoallelic expression to prevent false positives due to genotyping error. Here we repeat the same analysis with those SNPs included. We observe that removing SNPs with monoallelic expression (alt/total<0.02 or alt/total>0.98 in pseudobulk sample) have minimal effect on differential ASE except a small number of genes (< 1% for both DAESC-BB and DAESC-Mix) that switches from significant to insignificant, or vice versa (Supplementary Fig. 12). Second, we evaluate whether additional batch effects may confound the analysis. After adjusting for the month when the experiment was conducted, the number of discoveries and the validation rate virtually remain the same (Supplementary Data 3).

We further use the phased genotype data from Cuomo et al.[10] to validate the ability of DAESC-Mix to conduct implicit haplotype phasing. For 48.4% of the genes that reach significance (FDR < 0.05 by DAESC-Mix), DAESC-Mix learns two haplotype combinations with the minor haplotype including >10% individuals (Supplementary Fig. 13). We conduct the validation on the genes that show suggestive evidence of D-ASE by DAESC-Mix ($P < 0.05$) and have at least one eQTL reported by Cuomo et al.[10]. We further restrict to 179 genes that are significant according to a likelihood ratio test comparing DAESC-Mix to DAESC-BB (nominal $P < 0.05$). This restriction selects genes for which DAESC-Mix reports two haplotype combinations ($z_i = 1$ and $z_i = -1$). Fisher's exact test show that for 77 (43%) genes, the mixture labels given by DAESC-Mix successfully captures the observed haplotype combinations between the gene and the top eQTL ($P < 0.05$, Fig. 3f) according to phased data. An example was *NMU*, for which DAESC-Mix reports highly significant dynamic ASE ($p = 1.93 \times 10^{-59}$) and captures the haplotype combinations well ($p_{fisher} = 1.51 \times 10^{-6}$). We observe that allelic fractions move in opposite directions along pseudotime for two clusters of individuals, and combining two groups would severely diminish the apparent allelic effects (Fig. 3g). For 39 (22%) genes, mixture labels are not associated with haplotype combinations ($P > 0.5$). This could be due to imperfect eQTL calling by the original study, or limitations of our method. Due to its high power, high validation rate, and ability to capture haplotype combinations, we choose DAESC-Mix as the main method of discovery.

## Patterns and mechanisms of dynamic ASE

We hypothesize that dynamic ASE during differentiation could be linked to dynamic changes of chromatin state. To test this hypothesis, we use the chromatin states learned by ChromHMM[22] on the Roadmap Epigenomics data[15] (see "Methods" for details). We recode the chromatin states to 0 (inactive) and 1 (active) based on the criteria described in "Methods". For each gene, we compute the absolute value of change in chromatin state (0−inactive, 1−active) at the transcription start site (TSS) between two endpoints of differentiation: iPSC and definitive endoderm. The D-ASE genes identified by DAESC-Mix show an average chromatin state change of 0.132, while the non-D-ASE genes show an average change of 0.075 (Fig. 4a). This difference is highly significant even after adjusting for the read depth of the genes ($p = 3.19 \times 10^{-9}$). The D-ASE genes identified by DAESC-BB and the GLMM also show a larger change in chromatin state compared to non-D-ASE genes, but the difference is smaller, and for the GLMM (Fig. 4a). In addition, we observe significant correlations between the D-ASE effect size (log-OR when pseudotime changes from 0 to 1) and the magnitude of change in chromatin state, with DAESC-Mix showing the strongest correlation (Fig. 4b).

To further study the pattern of dynamic change in ASE, we compute the average allelic fraction for iPSCs and definitive endoderm cells using DAESC-Mix estimates (Methods). We find different genes

show allelic imbalance at different stages of differentiation (Fig. 4c). For example, genes *SFRP2* and *NMU* have minimal allelic imbalance at the iPSC stage but substantial imbalance at the definitive endoderm stage. On the contrary, genes *VIM* and *LEPREL1* only show allelic imbalance in iPSCs but not definitive endoderm cells. For genes *IFITM3*, *SNHG17* and *TRDN* the allelic imbalance appears at both stages of differentiation but with a different magnitude. Lastly, for genes *RAB17* and *GATM* the allelic fraction switches directions across stages, i.e., the highly expressed allele for iPSCs becomes the less expressed allele for definitive endoderm cells. To enable future exploration, we classify the 657 D-ASE genes identified by DAESC-Mix into six categories based on which differentiation stage shows allelic imbalance (Fig. 4d). See Methods for the classification criteria. More than half of the genes show stronger allelic imbalance in definitive endoderm cells than iPSCs (51.6% late and increasing, Fig. 4d), only 15.8% show stronger imbalance in iPSCs (early and decreasing, Fig. 4d).

As a validation analysis, we examine whether our top 30 D-ASE genes (Fig. 4c) have previously been reported to exhibit D-ASE, ASE, or other biological relevance in the literature. Moyerbrailean et al.[23] found that 23 out of the 30 genes have ASE in cell types, including lymphoblastoid cell lines (LCL), smooth muscle cells (SMC), murine erythroleukemia cells, HUVECs, and PBMCs. Fan et al.[14] reported 12 out of 30 genes have D-ASE in kidney, M0 macrophage cells, or M1 macrophage cells. Expression of some of the genes is tightly regulated in endodermic tissues. For example, *DKK1* was reported to be carefully regulated during kidney development;[24] *GSTO1* was shown to have ASE in mouse lung, liver, and brain;[25] and *GNAS* is a known imprinted gene in endodermal tissues such as pituitary[26], thyroid gland, and gonads[27]. Gene-set enrichment analysis find 121 Gene Ontology (GO) biological process gene sets enriched in D-ASE genes identified by DAESC-Mix, including those for the regulation of mesoderm development and cell development (Supplementary Data 2). In particular, the top 30 D-ASE genes identified by DAESC-Mix (Fig. 4c) are enriched in 10 GO biological processes gene sets (Fig. 4e). Most of the enriched gene sets are related to development or differentiation, including regulation of mesoderm development, dopaminergic neuron differentiation, cell fate specification, mesodermal cell differentiation, mesoderm formation, and gastrulation (Fig. 4e). This result validates the biological relevance of the D-ASE genes we discover.

## Type 2 diabetes and differential ASE in pancreatic islet cells

We obtain the scRNA-seq data from pancreatic islet samples of four type 2 diabetes (T2D) patients and six controls[8]. After preprocessing ("Methods"), we obtain single-cell ASE data for 2209 cells of 14 cell types (Fig. 5a, b). To identify genes potentially dysregulated in T2D patients, we conduct differential ASE analysis between cases and controls for four major endocrine cell types: alpha, beta, delta, and gamma cells. Due to the small sample size, we use DAESC-BB as the method for discovery. We find three genes that show differential ASE between cases and controls (FDR < 0.05, Fig. 5c). Differential ASE of *ARPC1B* and *SLC37A4* is only found in alpha cells, and differential ASE of *REEP5* is found in both alpha and beta cells. *SLC37A4* and *REEP5* show stronger allelic imbalance in T2D patients than controls (Fig. 5c), indicating that these regulatory effects are only present in T2D patients. *ARPC1B*, however, shows stronger allelic imbalance in healthy controls (Fig. 5c), indicating that the regulatory effects are potentially diminished in T2D patients. Among our hits, previous studies indicate a potential link between *SLC37A4* and T2D. *SLC37A4* encodes glucose 6-phosphate translocase, which transports glucose 6-phosphate from the cytoplasm to the endoplasmic reticulum[28,29]. rs7127212, which is 51.6 kb from the TSS of *SLC37A4*, was reported to be associated with the risk of T2D by a previous study[30]. Through this analysis, we demonstrate that DAESC can also detect differential ASE between case−control disease status, even when the data consist of only a few individuals.

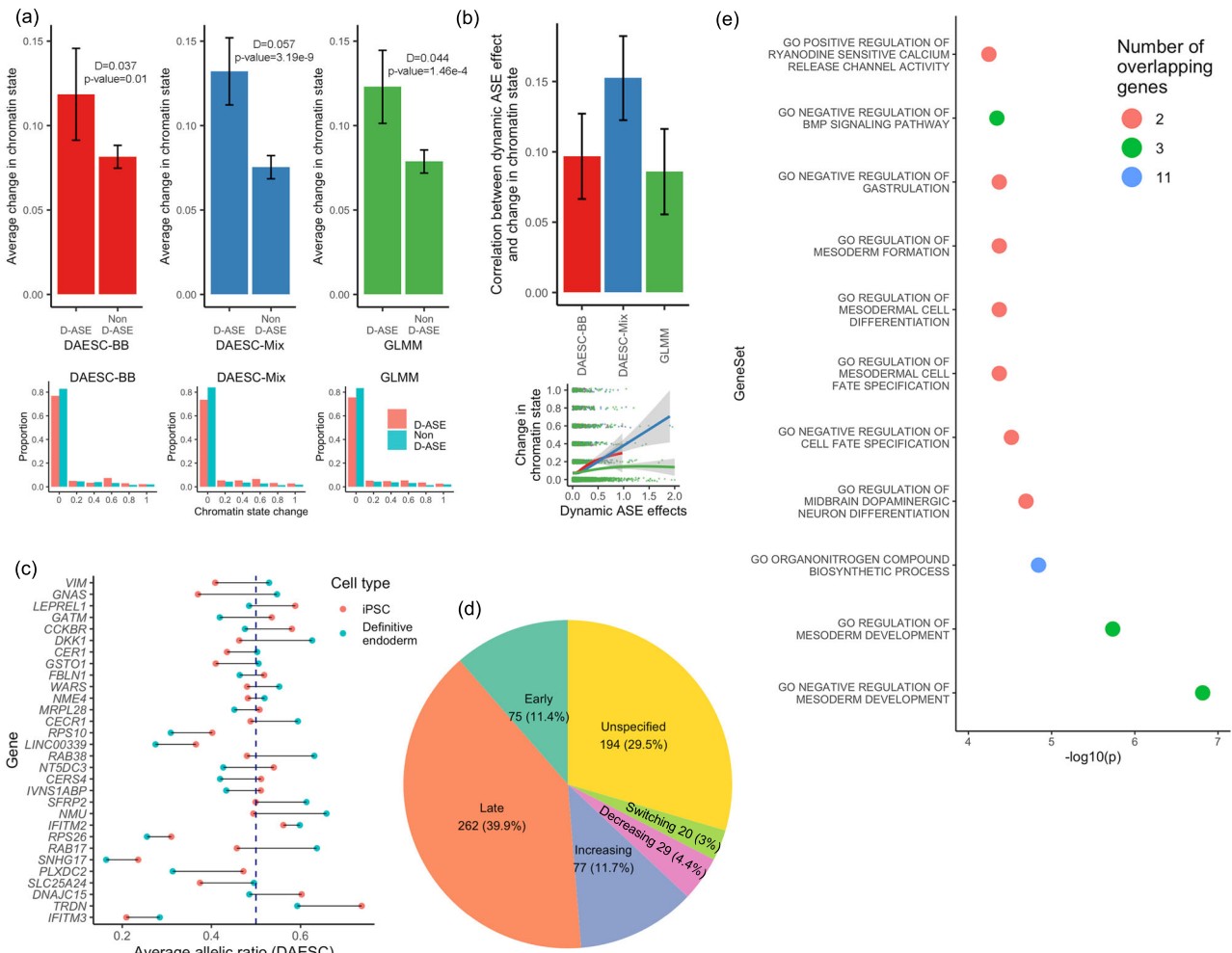

**Fig. 4 | Patterns and mechanisms of dynamic ASE genes during endoderm differentiation. a** Upper row: average change in chromatin state at transcription start site from iPSC to definitive endoderm cells for D-ASE genes and non-D-ASE genes; Error bars represent 95% confidence intervals (mean ± 1.96 standard error). Lower row: distribution of chromatin stage change (possible values: 0, 0.2, 0.4, 0.6, 0.8, 1) among D-ASE genes and non-D-ASE genes. At FDR < 0.05, the number of D-ASE genes is 324 for DAESC-BB, 657 for DAESC-Mix, and 532 for GLMM. For each method, the genes that do not reach FDR < 0.05 are considered non-D-ASE genes (see "Methods" for details). Chromatin states are from ChromHMM analysis of the Roadmap Epigenomics data and recoded to 0 (inactive) or 1 (active). D is the difference between D-ASE and non-D-ASE genes, and *P* values are calculated using linear regression: chromatin state change ~ I (the gene shows D-ASE) + total read depth of the gene. Here I(.) is the indicator

function. **b** Upper: correlation between D-ASE effect size ($\beta_1$) and change of chromatin state (*n* = 4102 genes). Error bars represent 95% confidence intervals (estimated correlation coefficient ± 1.96 standard error). Lower: scatter plot from which the correlation is derived; trend curves are generated using ggplot2::geom_smooth(), and shadings represent 95% confidence bands. **c** Top 30 genes identified by DAESC-Mix (smallest *P* values) and average allelic ratio of iPSCs vs definitive endoderm cells estimated by DAESC-Mix, computed as $1/(1 + \exp(-(\beta_0 + \beta_1 t)))$ where *t* is the average pseudotime of the cell type. The dashed line corresponds to allelic ratio=0.5. **d** Types of D-ASE genes and their proportions. See "Methods" for details. **e** Enrichment of top D-ASE 30 genes identified by DAESC-Mix in Gene Ontology Biological Processes gene sets. Only gene sets with FDR < 0.05 are shown. Source data are provided as a Source Data file.

## Discussion

Differential allele-specific expression is a powerful tool to study context-specific cis-regulatory effects. Single-cell RNA-seq (scRNA-seq) has allowed the study of ASE in heterogeneous cell types within a tissue. However, there is a lack of statistical tools for single-cell differential ASE analysis. In this paper, we describe DAESC, a generic statistical framework for differential ASE detection using scRNA-seq data from multiple individuals. The method captures sample repeat structure of multiple cells per individual using random effects, and DAESC-Mix further refines differential ASE analysis by incorporating implicit haplotype phasing. Simulation studies show that the method has well-controlled type I error and high power under a wide range of scenarios. Application to single-cell ASE data from an endoderm differentiation experiment identifies hundreds of genes that are dynamically regulated during differentiation. Dynamic regulatory effects are linked to changes in chromatin state at the TSS. The D-ASE genes

are enriched in GO terms related to development and differentiation. A second application to single-cell data from pancreatic islets identifies three genes with differential ASE between T2D patients and controls in alpha and beta cells, despite the small sample size.

Within the DAESC framework, the full model DAESC-Mix is generally more powerful than DAESC-BB. However, we recommend using DAESC-Mix when the number of individuals is reasonably large (e.g., $N \geq 20$), since the mixture model needs large *N* to identify different haplotype combinations. Indeed, simulation studies show that power gain was more pronounced under large *N* (Figs. 1 and 2 and Supplementary Figs. 3–5). When the sample size is small (e.g., *N*<20), the overall performance between DAESC-Mix and DAESC-BB is less distinguishable (see precision–recall curves in Supplementary Fig. 4). In that case, we recommend using DAESC-BB which has better type I error control. In our first application, the dataset from endoderm differentiation is comprised of 105 individuals and hence DAESC-Mix is

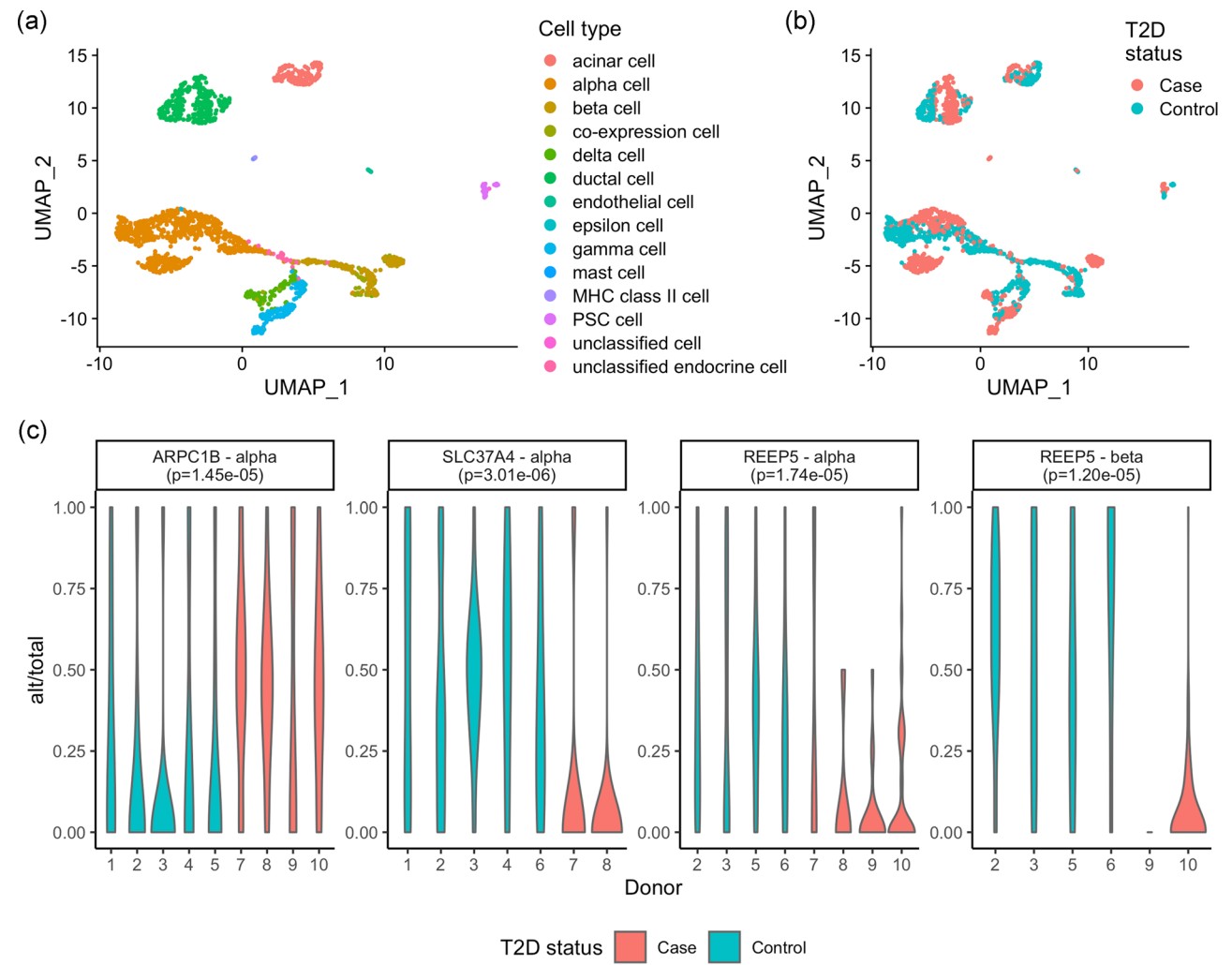

**Fig. 5 | Differential ASE between type 2 diabetes patients and controls in pancreatic endocrine cells.** UMAP colored by **a** cell type, **b** disease status. **c** Genes that show differential ASE between type 2 diabetes and controls in pancreatic endocrine cell types identified by DAESC-BB (FDR < 0.05, likelihood ratio test, two-sided) and distribution of allelic fraction in each donor. Alt: alternative allele read count; total: total allele-specific read count. P values in subgraph titles are original P values without multiple comparison adjustment. Source data are provided as a Source Data file.

chosen. In the second application, the pancreatic islet dataset is comprised of only 10 individuals, and hence DAESC-BB is chosen.

Note that the two-component mixture model used by DAESC-Mix is a simplifying assumption. When the gene has one eQTL, the true model should have an extra component corresponding to the individuals of whom the eQTL is homozygous. When the gene has multiple eQTLs, the number of true haplotypes grows exponentially. DAESC-Mix uses a two-component model to prevent overfitting and to increase computational speed. However, simulation studies show the performance of DAESC-Mix remains robust when there are multiple eQTLs (Fig. 2 and Supplementary Fig. 5). This is also due to the limitation of sample size, since the number of individuals in single-cell ASE datasets are often not enough to robustly fit a mixture model with many components. More complex mixture models may become viable as more data are collected, and could be extended from DAESC-Mix.

DAESC has important conceptual and technical differences from existing methods scDALI[11] and airpart[12]. First, DAESC is designed as a generic tool for differential ASE analysis with respect to any condition, regardless of whether the comparison is between cell-types within an individual or across individuals, and regardless of whether the condition of interest is continuous or discrete. The random effects that account for sample repeat structure is an important component that enables this flexibility. scDALI and airpart focus on differential ASE

across cell types, not across samples or individuals. They allow for adjustment of donor IDs as fixed effects but cannot be used for differential ASE across conditions between individuals (e.g., disease status). In contrast, GLMM fitted by lme4[31] is more comparable to DAESC-BB than scDALI or airpart. Both GLMM and DAESC use random effects to model sample repeat structure but they account for overdispersion differently. Therefore, GLMM is used as the main reference method for benchmarking and has similar precision–recall curve to DAESC-BB in some of the scenarios (Supplementary Fig. 4). Second, DAESC-Mix conducts implicit haplotype phasing to recover allelic signals hidden by haplotype switching. Hence DAESC-Mix can be powerful regardless of whether genotypes are available or eQTLs have been identified, which is not the case for many datasets. In the scDALI paper[11], the application to scRNA-seq data assigned the alternative haplotype of the gene based on the alternative allele of the eQTL. This approach is only possible if genotype data are available and if there is at least one significant eQTL for the gene. If the gene is regulated by multiple weak eQTLs that do not attain genome-wide significance, scDALI does not have a mechanism to assign alternative haplotypes. However, DAESC-Mix can still be used and may be able to capture the combined effects of multiple eQTLs as shown in the simulations (Fig. 2 and Supplementary Fig. 5). Previous methods for bulk RNA-seq have used a majority voting approach for pseudo haplotype phasing[5,14,32]. However,

this approach is not directly applicable to single-cell ASE due to multiple cells from each individual and low read depth per cell.

Our method does have some limitations to consider. First, we observed modest type I error inflation for DAESC-Mix potentially due to overfitting. However, the inflation seems acceptable given the magnitude of power improvement. If provided with enough computational resources, the users can choose to conduct permutation tests to further correct type I error. Second, DAESC-Mix is most powerful when applied to datasets with a large number of individuals, but such datasets are yet not widely available. For small datasets we recommend using DAESC-BB, which may be conservative but has well-controlled type I error. In the future, DAESC-Mix will be more widely applicable with the availability of larger datasets along with cheaper and better technology for large-scale single-cell profiling. Third, DAESC is not optimized for integrating information across multiple discrete cell types into a unified test. scDALI and airpart both have methods for this purpose. A future direction is to combine the strengths of DAESC and scDALI or airpart to incorporate sample repeat structure, implicit haplotype phasing and integration of information across cell types. A potential approach is to include cell types within individuals in a nested mixed model. In addition, DAESC is focused on statistical analysis post data processing. Given the complexity of ASE data processing, other factors such as variant calling approaches, quality control thresholds, sequencing read depth and platform can also have effects on the results. A comprehensive evaluation of the factors and an optimized analysis pipeline is an important area of research. Though this is beyond the scope of the paper, we demonstrate through simulations that the relative ranks of the methods are robust to the change of overdispersion and read depth (Supplementary Figs. 6 and 7), which are closely related to data quality. Lastly, DAESC is not specifically developed for analyzing cancer datasets. In particular, the implicit phasing in DAESC-Mix assumes every cell from an individual share the same genotype. This assumption is violated for cancer cells due to many somatic mutations. Single-cell ASE analysis in cancer cells is also an intriguing future direction.

In conclusion, we have developed a statistical method, DAESC, for efficient detection of differential ASE across any comparison of interest. Given the large volume of single-cell ASE data that are being generated, DAESC has great potential to facilitate the discovery of context-specific cis-regulatory effects in a wide range of scenarios.

# Methods
## DAESC model

We describe the DAESC model for differential ASE analysis using scRNA-seq data across multiple individuals. For a heterozygous tSNP, let $y_{ij}$ be the alternative allele read count for individual $i$ and cell $j$, and $n_{ij}$ be the total allele-specific read count. Let $x_{ij}$ be the independent variable, e.g., cell types, cell differentiation time, or disease status of the individual. Define $\mathbf{y}_i = (y_{i1}, \ldots, y_{iJ_i})$ where $J_i$ is the number of cells from individual $i$. DAESC is comprised of two components: a baseline beta-binomial regression model with individual-specific random effects (DAESC-BB), and a full beta-binomial mixture model that incorporates implicit haplotype phasing (DAESC-Mix).

The DAESC-BB model is formulated as follows:

$$y_{ij}|n_{ij} \sim BB(n_{ij}, \mu_{ij}, \phi)$$
$$\log\left(\frac{\mu_{ij}}{1-\mu_{ij}}\right) = \beta_0 + \beta_1 x_{ij} + a_i \tag{1}$$
$$a_i \sim N(0, \sigma_a^2)$$

Here $BB(n_{ij}, \mu_{ij}, \phi)$ is a beta-binomial distribution with denominator $n_{ij}$, mean proportion $\mu_{ij}$ and overdispersion parameter $\phi$. It is equivalent to $y_{ij}|n_{ij} \sim binomial(n_{ij}, p_{ij})$, $p_{ij} \sim beta\left(\frac{\mu_{ij}}{\phi}, \frac{1-\mu_{ij}}{\phi}\right)$ marginalized over $p_{ij}$. We model $\log\left(\frac{\mu_{ij}}{1-\mu_{ij}}\right)$ as a linear function of $x_{ij}$. The individual-specific random effect $a_i$ accounts for the sample repeat structure introduced

by having multiple cells from each individual. This model can be used for any differential ASE analysis but may be conservative in some scenarios due to unknown causal variants and haplotype information. For example, when the tSNP is not in strong LD with the causal eQTL, different individuals may exhibit complementary allelic fractions which actually reflect the same regulatory effect. Failing to account for this possibility can lead to diminished ASE signal when aggregated across individuals.

This issue can be addressed using DAESC-Mix when the sample size (number of individuals) is sufficiently large. The model is formulated as follows:

$$y_{ij}|n_{ij} \sim BB(n_{ij}, \mu_{ij}, \phi)$$
$$\log\left(\frac{\mu_{ij}}{1-\mu_{ij}}\right) = z_i(\beta_0 + \beta_1 x_{ij}) + a_i$$
$$z_i = 2\delta_i - 1, \delta_i \sim \text{Bernoulli}(\pi_0) \tag{2}$$
$$a_i \sim N(0, \sigma_a^2)$$

This model is an extension of DAESC-BB with the inclusion of an indicator variable $z_i$. It models the scenario where ASE is caused by one eQTL. When $z_i = 1$, the alternative allele of the eQTL and the alternative allele of the transcribed SNP (tSNP) are on the same haplotype, and the reference alleles of the two SNPs are on the same haplotype. When $z_i = -1$, the alternative allele of the eQTL and the reference allele of the tSNP are on the same haplotype, and vice versa (Fig. 1). Though it is possible that the eQTL is homozygous for some individuals, we do not model this scenario to prevent overfitting and speed up computation.

Though the models above are described for a heterozygous tSNP, it can also be applied to gene-level ASE counts generated by aggregating across multiple tSNPs.

## Model inference by variational EM

The inference is conducted by a variational EM algorithm[33]. Here we describe the algorithm for DAESC-Mix. Details of the derivation and the algorithm for DAESC-BB can be found in Supplementary Notes. Denote $\boldsymbol{\beta} = (\beta_0, \beta_1)^T$. We treat $a_i$ and $\delta_i$ as missing data and the complete data likelihood is

$$P(\mathbf{y}_1, a_1, \delta_1, \ldots, \mathbf{y}_N, a_N, \delta_N | \boldsymbol{\beta}, \sigma_a^2, \phi, \pi_0)$$
$$= \prod_i P(\mathbf{y}_i, a_i, \delta_i | \boldsymbol{\beta}, \sigma_a^2, \phi, \pi_0)$$
$$\propto \prod_i \left\{ \pi_0 \prod_j \frac{B\left(\frac{\mu_{ij1}}{\phi} + y_{ij}, \frac{1-\mu_{ij1}}{\phi} + n_{ij} - y_{ij}\right)}{B\left(\frac{\mu_{ij1}}{\phi}, \frac{1-\mu_{ij1}}{\phi}\right)} \right\}^{\delta_i} \tag{3}$$
$$\times \left\{ (1 - \pi_0) \prod_j \frac{B\left(\frac{\mu_{ij2}}{\phi} + y_{ij}, \frac{1-\mu_{ij2}}{\phi} + n_{ij} - y_{ij}\right)}{B\left(\frac{\mu_{ij2}}{\phi}, \frac{1-\mu_{ij2}}{\phi}\right)} \right\}^{1-\delta_i} (\sigma_a^2)^{-\frac{1}{2}} \exp\left(-\frac{a_i^2}{2\sigma_a^2}\right)$$

Here $\mu_{ij1} = \frac{\exp(\beta_0 + \beta_1 x_{ij} + a_i)}{1 + \exp(\beta_0 + \beta_1 x_{ij} + a_i)}$ and $\mu_{ij2} = \frac{\exp(-(\beta_0 + \beta_1 x_{ij}) + a_i)}{1 + \exp(-(\beta_0 + \beta_1 x_{ij}) + a_i)}$. The variational EM iteration goes as follows:

In the E-step, we use variational inference[34,35] to approximate the posterior distribution $P\left(a_i, \delta_i | \mathbf{y}_i, \boldsymbol{\beta}_{(t)}, \sigma_{a,(t)}^2, \phi_{(t)}\right)$, where $\boldsymbol{\beta}_{(t)}, \sigma_{a,(t)}^2, \phi_{(t)}$ are the parameter values at iteration $t$. We use the mean field approximation $q(a_i, \delta_i) = q(a_i)q(\delta_i)$ with a delta method approximation[34]. Denote the variational distribution by

$$q(a_i) = N\left(\hat{a}_{i,(t)}, \hat{\sigma}_{a_i,(t)}^2\right), q(\delta_i) = \text{Bernoulli}\left(\pi_{i,(t)}\right). \tag{4}$$

See Supplementary Notes for details of the derivation.

In the M-step, we first update $\pi_0$ by $\pi_{0,(t+1)} = \frac{1}{N}\sum_i \pi_{i,(t)}$ and update $\sigma_a^2$ by $\sigma_{a,(t+1)}^2 = \frac{1}{N}\sum_i \hat{a}_{i,(t)}^2 + \hat{\sigma}_{a_i,(t)}^2$. Update $\boldsymbol{\beta}$ and $\phi$ by numerical

optimization of the following objective function:

$$Q\left(\boldsymbol{\beta}, \boldsymbol{\phi} \mid \boldsymbol{\beta}_{(t)}, \boldsymbol{\phi}_{(t)}\right) = \sum_i E_{q(a_i,\delta_i)} \left\{ \log P\left(\mathbf{y}_i, a_i, \delta_i \mid \boldsymbol{\beta}, \sigma^2_{a,(t)}, \boldsymbol{\phi}\right) \right\}. \quad (5)$$

Here $E_{q(a_i,\delta_i)}\{\cdot\}$ is the expectation under variational distribution $q(a_i, \delta_i)$.

After the parameter estimation, we test the null hypothesis $H_0$: $\beta_1 = 0$ using likelihood ratio test. Rejecting this null hypothesis indicates that there is differential ASE with respect to the independent variable $x_{ij}$. The method is implemented in R package DAESC (see Code Availability). Simulation studies and data analyses are conducted using R/4.0.2.

## Simulation studies

We conduct simulation studies using total read counts and parameters estimated from a real endoderm differentiation dataset[10]. The dataset is comprised of 4102 genes and 30,474 cells collected from 105 donors. See Methods subsection "Single-cell ASE data from endoderm differentiation" for details of the study. We randomly select 3000 genes and used the real total allele-specific read counts as the total allele-specific read counts ($n_{ij}$) in our simulations. This setting reflects realistic read depth and number of cells, but does not affect ASE which depends on the relative abundance of reference and alternative alleles. We simulate the alternative allele read counts assuming that there is only one eQTL driving ASE

$$y_{ij} \mid n_{ij} \sim BB(n_{ij}, \mu_{ij}, \phi)$$

$$\log\left(\frac{\mu_{ij}}{1-\mu_{ij}}\right) = z_i(\beta_0 + \beta_1 x_{ij} + \beta_1 \eta_i) + a_i \quad (6)$$

$$a_i \sim N(0, \sigma_a^2), z_i \sim \text{categorical}([-1,1,0], [\pi_1, \pi_2, \pi_3])$$

In contrast to the DAESC-Mix model, this simulation model introduces a third possible value of the latent variable $z_i$. Besides two values −1 and 1 which are modeled by DAESC-Mix, the third value $z_i = 0$ corresponds to the individuals for which the eQTL SNP is homozygous. The haplotype proportions $\pi_1, \pi_2, \pi_3$ are simulated based on given LD coefficient ($r^2$) between the eQTL and tSNP (see Supplementary Notes for details). We vary $r^2$ to 0, 0.1 and 0.9, and simulate 1000 genes for each value of $r^2$ including 500 null genes and 500 non-null genes.

We include two covariates in the simulation to evaluate the performance of DAESC under two types of D-ASE. The continuous covariate $x_{ij}$ is the real pseudotime provided by the original study;[10] the discrete covariate $\eta_i$ is a simulated sample-level disease status which can take values 0 or 1. A randomly chosen half of the individuals are assigned $\eta_i = 0$ (control) and the other half are assigned $\eta_i = 1$ (case).

To choose realistic values of other parameters, we apply DAESC-BB to the real data and obtain estimates of $\beta_0$, $\beta_1$, $\sigma_a^2$ and $\phi$. We select the genes with top 500 largest $|\beta_1|$ as potential values of parameters for the simulation. For each of the 3000 genes, we randomly select a set of parameters ($\beta_0, \beta_1, \sigma_a^2, \phi$) from the 500 candidate sets of values. For null genes we reset $\beta_1 = 0$. The 500 sets of candidate values are provided in Supplementary Data 4 distribution of the parameters is visualized in Supplementary Fig. 14.

We also vary the sample size to $N = 10, 50, 100$. For D-ASE with respect to $x_{ij}$, we randomly sample $N$ individuals from the simulated data for D-ASE with respect to $\eta_i$, we randomly sample $N/2$ cases and $N/2$ controls. We repeat this procedure 10 times and obtain 5000 simulations for each scenario (combination of $N$, $r^2$, differential ASE status). We observe the minimal variation of type I error and power across 10 replications (Supplementary Fig. 15). For the rest of the simulation studies, we conduct 400 simulations for each scenario to save computational time.

To avoid any bias toward the beta-binomial model, we conduct another simulation using the binomial GLMM. The simulation model is similar to the beta-binomial model except that overdispersion is generated by a cell-specific random effect ($\epsilon_{ij}$) instead of the beta distribution.

$$y_{ij} \mid n_{ij} \sim \text{Binomial}(n_{ij}, \mu_{ij})$$

$$\log\left(\frac{\mu_{ij}}{1-\mu_{ij}}\right) = z_i(\beta_0 + \beta_1 x_{ij} + \beta_1 \eta_i) + a_i + \epsilon_{ij} \quad (7)$$

$$a_i \sim N(0, \sigma_a^2), \epsilon_{ij} \sim N(0,1), z_i \sim \text{categorical}([-1,1,0], [\pi_1, \pi_2, \pi_3])$$

Parameters $\beta_0$, $\beta_1$, $\sigma_a^2$, $\pi_1$, $\pi_2$, $\pi_3$ are generated using the same procedure as the beta-binomial simulation.

## Simulations with multiple eQTL SNPs per gene

Due to the large number of scenarios for LD among eQTLs and the tSNP, we conduct this simulation study under a simplified scenario: all the eQTLs are independent from each other and independent from the tSNP. Similar to the one-eQTL scenario, we simulate the data using beta-binomial mixture model. Because the number of mixture components grow with the number of eQTLs, we simulate the mixture components indirectly by simulating the genotypes of the eQTLs. The steps are as follows:

- Randomly choose ($\sigma_a^2, \phi$) from 500 sets of candidate values (Supplementary Data 4). Parameters ($\sigma_a^2, \phi$) are the same across all mixture components.
- Simulate the minor allele frequency (MAF) of $m$ eQTLs, from $MAF_1, MAF_2, \ldots, MAF_m \sim \text{Uniform}[0.1, 0.5]$.
- Simulate the alleles of eQTLs that resides on the haplotype of the <u>reference</u> allele of the tSNP for N individuals, denoted by $g_{ik0} \sim bernoulli(MAF_k)$, $i = 1, \ldots, N; k = 1, \ldots, m$.
- Simulate the alleles of eQTLs that resides on the haplotype of the <u>alternative</u> allele of the tSNP, denoted by $g_{ik1}$, $i = 1, \ldots, N; k = 1, \ldots, m$.
- Draw $m$ pairs of regression coefficients ($\beta_0, \beta_1$) from 500 candidate sets of values (Supplementary Data 4), denoted by ($\beta_{10}, \beta_{11}$), $\ldots$, ($\beta_{m0}, \beta_{m1}$).
- Compute individual-specific ASE effects size as $\beta_{i0}^{ASE} = \sum_{k=1}^m \beta_{k0}(g_{ik1} - g_{ik0})$, $\beta_{i1}^{ASE} = \sum_{k=1}^m \beta_{k1}(g_{ik1} - g_{ik0})$.
- Compute $\mu_{ij}$ from $\log(\frac{\mu_{ij}}{1-\mu_{ij}}) = \beta_{i0}^{ASE} + \beta_{i1}^{ASE} x_{ij} + \beta_{i1}^{ASE} \eta_i + a_i$. For individuals who have the same set of $g_{ik1} - gi_{k0}$ ($k = 1, \ldots, m$), $\beta_{i0}^{ASE}$ and $\beta_{i1}^{ASE}$ are the same and hence the model collapses into the beta-binomial mixture model.
- Generate $y_{ij} \sim BB(n_{ij}, \mu_{ij}, \phi)$.

We vary the number of eQTLs to $m = 2, 3, 4, 5, 6$.

## Other methods for comparison

We compare DAESC-BB and DAESC-Mix to other methods: GLMM, apeglm, apeglm-adj, EAGLE, and EAGLE-PB.

The first method is a generalized linear mixed model (GLMM) implemented by the lme4 package in R. The GLMM is formulated as follows:

$$y_{ij} \mid n_{ij} \sim Binomial(n_{ij}, p_{ij})$$

$$\log\left(\frac{p_{ij}}{1-p_{ij}}\right) = \beta_0 + \beta_1 x_{ij} + a_i + \epsilon_{ij} \quad (8)$$

$$a_i \sim N(0, \sigma_a^2), \epsilon_{ij} \sim N(0, \sigma_\epsilon^2)$$

The R formula is cbind(y,n-y) ~ x + (1|subj) + (1|obs), where subj is the individual ID and obs is the unique ID for each cell. Here $a_i$ accounts for sample repeat structure and $\epsilon_{ij}$ accounts for overdispersion.

Apeglm is a fixed-effects beta-binomial regression:

$$y_{ij}|n_{ij} \sim BB\left(n_{ij}, \mu_{ij}, \phi\right), \log\left(\frac{\mu_{ij}}{1 - \mu_{ij}}\right) = \beta_0 + \beta_1 x_{ij} \qquad (9)$$

This model does not account for the sample repeat structure of single-cell ASE data. Therefore, we include a variation of apeglm (apeglm-adj) into the comparison, which further adjusts for donor IDs as fixed-effects covariates. Note that apeglm-adj can only be used for differential ASE with respect to a continuous variable but not binary case–control status, which is colinear with the one-hot encoding of donor IDs.

EAGLE[5] is another method developed for differential ASE analysis using bulk RNA-seq data. We first apply EAGLE directly to single-cell ASE data without accounting for the sample-repeat structure. For differential ASE across disease status, we further compare with EAGLE applied to pseudobulk data (EAGLE-PB). We aggregate cells from each individual into a pseudobulk sample by summing the alternative and total read counts. We then apply EAGLE to test for differential ASE using the pseudobulk samples.

## Single-cell ASE data from endoderm differentiation

Cuomo et al.[10] conducted an endoderm differentiation experiment of 125 induced pluripotent stem cell (iPSC) lines from the Human Induced Pluripotent Stem Cell initiative (HipSci). Gene expression was profiled at 4 differentiation times points using single-cell RNA-seq (Smart-seq2). We obtain SNP-level allele-specific read counts for 114 donors from (https://zenodo.org/record/3625024#.YnJ-ivPMKi4), and restrict to 105 individuals for which genotype data are available to us. We remove SNPs with low mappability (ENCODE 75-mer mappability <1), and those with monoallelic expression to reduce the effect of potential genotyping error. Monoallelic expression is defined for each SNP in each individual by ALT/TOTAL < 0.02 or ALT/TOTAL > 0.98[20], where ALT is the sum of alternative allele read counts for all cells from the individual, and TOTAL is the corresponding sum of total allele-specific read counts.

## Aggregating SNP-level ASE counts to gene-level

Since phased genotype data are needed to aggregate SNP-level ASE counts to gene-level ASE counts, we impute and phase the genotype data using the Michigan Imputation Server with the Haplotype Reference Consortium (HRC) r1.1 data as the reference panel. For each individual and each gene, we sum the ASE counts across all SNPs within the exonic regions of the gene for each haplotype and obtain two haplotype-specific counts (hap1 count and hap2 count). Coordinates of exonic regions are provided by GTEx v7[36] annotation files (hg19) based on collapsed gene model. After removing the genes which had non-zero ASE counts in ≤20% of the cells, we obtain ASE counts for 4102 genes and 30,474 cells.

For joint analysis across individuals, alternative and reference haplotypes need to be consistently assigned across individuals. In the paper by Cuomo et al.[10]., the haplotype which is on the same chromosome as the alternative allele of the eQTL is assigned as the alternative haplotype. However, we would like to conduct ASE analysis without calling eQTL first, as is the case in many other studies. Therefore, we assign alternative and reference haplotypes based on the tSNP which has the highest total allele-specific read count across individuals (referred to by top tSNP), i.e., the haplotype on the same chromosome as the alternative allele of the top tSNP is assigned as the alternative haplotype. For those individuals for which the top tSNP is homozygous, alternative and reference haplotypes were assigned randomly.

## Validation of differential ASE genes

The list of dynamic eGenes reported by Cuomo et al.[10] can be used to validate our dynamic ASE findings. Since dynamic ASE is aimed to

capture dynamically regulation of gene expression, dynamic ASE genes should have substantial overlap with dynamic eGenes. Therefore, we compare the proportion of significant dynamic ASE (FDR < 0.05) that overlap with dynamic eGenes. To alleviate any doubt that different validation rates are caused by different numbers of genes identified by the methods, we create a concordance-on-top plot to compare the same number of top genes for all methods, which is varied from 10 to 800.

## Comparing DAESC-Mix mixture labels and observed haplotype combinations

Since phased genotype data are available for this study, we can use them to validate the ability of DAESC-Mix to capture haplotype combinations. For each gene, we obtain a posterior probability ($p_{mix}$) for each individual to belong to the first group. We assign the individual to the first group if $p_{mix} > 0.5$, or the second group if $p_{mix} < 0.5$. To compare with observed haplotype combinations, we first identify the top eQTL reported by Cuomo et al. for each of the genes above. The original paper identified eQTL for three cell types separately: iPSC, mesendoderm cells and definitive endoderm cells. We choose the SNP that shows the strongest association $P$ value in any of the three cell types as the top eQTL for the gene. There are three possible observed haplotype combinations: (1) $alt_{eQTL}, alt_{gene}|ref_{eQTL}, ref_{gene}$, (2) $alt_{eQTL}, ref_{gene}|ref_{eQTL}, alt_{gene}$, (3) $alt_{eQTL}, alt_{gene}|alt_{eQTL}, ref_{gene}$ or $ref_{eQTL}, alt_{gene}|ref_{eQTL}, ref_{gene}$. Here $ref_{eQTL}$ and $alt_{eQTL}$ are the reference and alternative alleles of the top eQTL, respectively; $ref_{gene}$ and $alt_{gene}$ are the reference and alternative haplotypes of the gene, respectively. Alleles or haplotypes on same side of "|" are on the same haplotype. We tally the number of individuals in two mixture groups vs. three haplotype combinations into a 2 × 3 table (Fig. 3). Finally, we perform Fisher's exact test on the 2 × 3 table to test the association between mixture clusters and observed haplotype combinations.

## Dynamic eGene clustering

We explore the total expression trends of (1) previously discovered dynamic eQTL genes by Cuomo et al.[10] and (2) the set of dynamic ASE genes discovered using DAESC-Mix (Supplementary Data 1). Pseudo-time smoothing is performed as in Cuomo et al.[10], and spectral clustering is performed on pseudotime-smoothed total expression using Pearson correlation as the affinity metric. In order to maintain a meaningful comparison with the original analysis, four clusters are used for both analyses.

## Chromatin-state analysis

We download the chromatin states learned by ChromHMM[22] for the Roadmap Epigenomics Project[15] (https://egg2.wustl.edu/roadmap/web_portal/chr_state_learning.html). For each gene, we compare the chromatin state at the TSS between iPSCs and endoderm cells. We consider chromatin states ≤ 7 as active, including 1_TssA, 2_TssAFlnk, 3_TxFlnk, 4_Tx, 5_TxWk, 6_EnhG, and 7_Enh, and assign them value 1 to represent active states in general. The remaining states are considered inactive and assigned value 0. Since there are multiple epigenomics for iPSCs (E018-E022, https://docs.google.com/spreadsheets/d/1yikGx4MsO9Ei36b64yOy9Vb6oPC5IBGlFbYEt-N6gOM/edit#gid=15), we use the average chromatin states (0 to 1) as the chromatin state for iPSC. We then compute the absolute difference of chromatin state between iPSC vs. hESC-derived CD184+ endoderm cultured cells (E011), which we refer to as chromatin state change.

For three D-ASE methods, DAESC-BB, DAESC-Mix and the GLMM, we compute the average chromatin state change for D-ASE genes (FDR < 0.05) and non-D-ASE genes (FDR ≥ 0.05), respectively. There are 324 D-ASE genes and 3,778 non-D-ASE genes identified by DAESC-BB, 657 D-ASE genes and 3,445 non-D-ASE genes identified by DAESC-Mix, and 1,995 D-ASE genes and 2,107 non-D-ASE genes identified by the GLMM. To test the significance of the difference between D-ASE

and non- D-ASE genes, we use linear regression adjusting for the total number of allele-specific reads for each gene: chromatin state change -I(D-ASE) + total read depth of the gene. This adjustment removes the effect of total expression, which can be a potential confounder. We also compute the correlation between D-ASE effect size ($\beta_1$) and chromatin-state change.

## Gene-set enrichment

We conduct gene set enrichment analysis for 657 D-ASE genes identified by DAESC-Mix using FUMA GWAS[37]. We only consider Gene Ontology (GO) biological process pathways[38] and use protein-coding genes as background. Finally, gene sets with FDR-adjusted enrichment *P* value < 0.05 are considered as significantly enriched.

## Classification of dynamic ASE genes

We classify the D-ASE genes identified by DAESC-Mix based on the stage of differentiation where allelic imbalance occurs. For each D-ASE gene, we first compute the average allelic fraction for iPSCs ($p_{ipsc}$) and definitive endoderms ($p_{defendo}$) estimated by DAESC-Mix as $1/(1 + \exp(-(\beta_0 + \beta_1 t)))$, where $t$ is the average pseudotime of the cell type. See Cuomo et al.[10] for the classification of cell types. Genes are classified into the following categories based on their ASE patterns:

- Increasing: $p_{defendo} < p_{ipsc} < 0.47$ or $p_{defendo} > p_{ipsc} > 0.53$.
- Decreasing: $p_{ipsc} < p_{defendo} < 0.47$ or $p_{ipsc} > p_{defendo} > 0.53$.
- Late: $|p_{ipsc} - 0.5| < 0.03$ and $|p_{defendo} - 0.5| > 0.03$
- Early: $|p_{ipsc} - 0.5| > 0.03$ and $|p_{defendo} - 0.5| < 0.03$
- Switching: $p_{ipsc} < 0.47$ and $p_{defendo} > 0.53$, or $p_{defendo} < 0.47$ and $p_{ipsc} > 0.53$

Other genes are classified as unspecified.

## Pancreatic islet data

Segerstolpe et al.[8] collected scRNA-seq data from pancreatic islet samples of four type 2 diabetes (T2D) patients and six controls. Libraries were prepared using Smart-seq2 protocols and sequencing was conducted using single-end 43 bp reads. We download raw fastq files from ArrayExpress and trimmed the reads with trimmomatic v0.38[39]. Reads are aligned to hg19 reference genome using STAR 2.7.10a[40]. Duplicated reads are marked with Picard 2.18.

Before obtaining ASE counts call, we first call genetic variants from scRNA-seq data using GATK (4.0.0). We follow the GATK best practices workflow for RNAseq short variant discovery. After further preprocessing steps (SplitNCigarReads and base recalibration), we merge the bam files of all cells from each individual into a pseudobulk bam file per individual. We then call variants using GATK HaplotyperCaller with the ten pseudobulk bam files as input. We extract biallelic SNPs from the called variants. We then obtain single-cell ASE counts using GATK ASEReadCounter. We only retain the 2,209 cells that passed quality in the original paper[8] and discard the rest.

For each individual, we remove SNPs with potential genotyping error. Specifically, we remove SNPs with genotyping read depth ≤10 and genotyping quality ≤15. We further remove the SNPs with mono-allelic expression, defined by pseudobulk allelic fraction <0.05 or >0.95. The pseudobulk allelic fraction is defined as $\frac{\text{sum of alternative allele counts}}{\text{sum of total allele-specific counts}}$, where the sums are taken across cells from the individual. The purpose of this step is to further remove genotyping error.

To reduce the effect of alignment errors, we remove the SNPs with ENCODE 40-mer mappability <1. We then aggregate ASE counts from SNP level to gene level using a pseudo phasing approach used by the ASEP paper[14]. This pseudo-phasing approach is performed on four major endocrine cells: alpha, beta, gamma, and delta cells. We aggregate ASE counts from these four cell types into pseudobulk ASE counts. If there are multiple heterozygous tSNP within a gene, we sum the counts for the expression minor allele (the one with lower allele-

specific read count) of all tSNPs as the alternative haplotype read count for the gene.

For cell-type-specific differential ASE analysis, we only analyze genes for which ASE counts are available for a reasonably large number of cells and individuals. For each gene, we first remove individuals with <3 cells or <5 reads from the cell type. We drop the gene from D-ASE analysis if there are <50 cells or <2 cases or <2 controls remaining.

## URLs

HipSci: https://www.hipsci.org/

ArrayExpress: https://www.ebi.ac.uk/arrayexpress/

ENCODE mappability: https://genome.ucsc.edu/cgi-bin/hgFileUi?db=hg19&g=wgEncodeMapability

Trimmomatic: http://www.usadellab.org/cms/?page=trimmomatic

STAR: https://github.com/alexdobin/STAR

Picard: https://broadinstitute.github.io/picard/

GATK: https://gatk.broadinstitute.org/hc/en-us

GATK Best Practices Workflows: https://gatk.broadinstitute.org/hc/en-us/sections/360007226651-Best-Practices-Workflows.

## Reporting summary

Further information on research design is available in the Nature Portfolio Reporting Summary linked to this article.

## Data availability

The ASE data, cell metadata, and gene expression from endoderm differentiation are available at https://zenodo.org/record/3625024#.YnJ-ivPMKi4. HipSci genotype data used in this study are available via https://www.hipsci.org/lines/#/files?Assay%5B%5D=Genotyping%20array. The pancreatic islet data are available on ArrayExpress via accession number E-MTAB-5061. GENCODE hg19 reference genome is available via https://www.gencodegenes.org/human/release_44lift37.html. Source data are provided with this paper.

## Code availability

The DAESC R package and other analysis scripts are available on GitHub:[41] https://github.com/gqi/DAESC. A step-by-step tutorial of the analytical pipeline is available at https://github.com/gqi/DAESC/wiki.

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

## Acknowledgements

We thank Radhika Jangi for feedback on the manuscript. H.J. is supported by NIH/NHGRI grants R01HG009518 and R01HG010889. A.B. is supported by NIH/NIGHMS award R35GM139580 and the Chan Zuckerberg Initiative.

## Author contributions

G.Q., B.J.S., and A.B. conceived the idea. A.B. supervised the project. G.Q. developed the methods and conducted the simulation studies and data analyses. J.M.P. and R.K. conducted part of the functional follow-up analyses of endoderm differentiation data. H.J. and A.B. provided feedback on statistical methods and analyses. G.Q. drafted the manuscript. G.Q., R.K., and A.B. edited the manuscript. All authors reviewed the manuscript.

## Competing interests

A.B. consults for Third Rock Ventures, Inc, and is a shareholder in Alphabet, Inc. The remaining authors declare no competing interests.
