## [Peer Review File · Nature Communications]

Single-cell allele-specific expression analysis reveals dynamic and cell-type-specific regulatory effectsReviewers' Comments:

Reviewer #1:

Remarks to the Author:

This paper proposes a differential ASE analysis tool that considers two elements not examined by existing approaches and I consider it to be a very fascinating method. Despite the authors' mention of data size as a limitation, I believe that the current trend of scRNA-seq becoming more cost-effective will result in widespread adoption of this tool. The tool enables numerous analyses based on covariate variables such as cell type and trajectory, thereby enhancing its significance. The paper is well-written, and the data demonstrate the method's validity. Overall, the manuscript is well-written and can be published with minor revisions.

- 1) The perspective of the dispersion parameter was not discussed. In the case of single-cell RNA-seq, data quality issues such as zero-inflation can be considerable, and data quality is likely to influence the dispersion parameter. In general, sample size and variance parameter have an inverse relationship in terms of statistical power. I ask for comments on expected results regarding the data quality. For example, based on the simulation study, will glmm be superior when data quality is poor?
- 2) Experimental species determine the appropriate sample size in experimental design. In the analysis of mouse data with low variance, a sample size of 5 is already substantial. From this perspective, it would be beneficial to include species-specific guidelines or comments for the method.
- 3) In Figure 1 and 2, A) -> a), which is not consistent with Figure 4. a)
- 4) In Figure 1, it would be better to relocate, either to top or bottom, the graph legend of c) and d).
- 5) In Figure 3-f, count -> Count in y-axis.

Reviewer #2:

Remarks to the Author:

The manuscript introduces a novel statistical method, DAESC, for differential allele-specific expression (ASE) analysis using single-cell RNA sequencing (scRNA-seq) data from multiple individuals. DAESC accounts for non-independence between cells from the same individual and incorporates implicit haplotype phasing. The method successfully identified dynamically regulated genes during endoderm differentiation and differentially regulated genes between type-2 diabetes patients and controls in pancreatic endocrine cells, providing new insights into gene regulation.

Here are my comments:

1. It would be beneficial to provide a comprehensive tutorial outlining the complete analytical pipeline of scRNA-seq ASE analysis, from raw data to DAESC analysis. This should include steps such as GATK, pseudo-bulk, etc.
2. Although the authors demonstrated the robustness and superiority of DAESC over GLMM, the potential impact of preprocessing and upstream operations on their approach has not been evaluated. Factors such as different variant calling approaches, thresholds, scRNA-seq read depth, library size, 3' or 5', and platform differences should be systematically assessed and discussed. A systematic evaluation and an optimized, comprehensive analytical pipeline would be highly appreciated.
3. The simulation study comparing DAESC to GLMM seems biased to me, as it employs the same distribution (beta-binomial) as the authors' model. It would be valuable to investigate whether DAESC can outperform (or at least be comparable to) GLMM in simpler binomial cases.
4. The comparison to only a GLMM model is limited, as other scRNA-seq and bulk data-based models have been developed. The authors should compare DAESC to a broader range of approaches in an

unbiased manner.

5. Applying DAESC to cancer datasets, which typically exhibit more genomic changes and potential eQTLs, would demonstrate the versatility of the method and increase its relevance to the cancer research community.

6. The authors model different cell types as independent variables. It would be interesting to explore whether a nested mixed model, with different cell types nested within individuals, could also be effective.

7. Combining the DAESC-BB and DAESC-Mix models into a single function with an automatic switch or a hyperparameter for model selection could make the method more user-friendly and cohesive.

8. The haplotype phasing in the DAESC-Mix model using EM is a novel aspect of the paper. However, the accuracy of the EM estimation and the validation of the results should be addressed. Additionally, the ratio of $z=1$ and $z=-1$ estimated in real-world datasets should be discussed (considering 1 could be more common than -1).

9. In many simulation studies, DAESC-BB shows similar results to GLMM in PR curves. This observation should be discussed and explained.

10. The authors should conduct a larger number of simulations and report error bars (or confidence intervals) for the performance metrics.

11. The authors should investigate the computational time of DAESC across varying numbers of cells and patients in scRNA-seq data.

12. The potential impact of single-cell data quality, library size, and batch effects on the analysis should be discussed.

13. While the allelic imbalance findings across cell differentiation in Figure 4 are intriguing, the authors should provide related studies to validate their observations.

14. Validation strategies for the identified genes and their roles in cell differentiation and disease development should be discussed, as well as the reliability and robustness of the results.

15. To further support the findings from real-world data, the authors should compare their results to those obtained using other approaches, such as scDALI and airpart. This will help alleviate any concerns regarding the validity of the results.

Reviewer #3:

Remarks to the Author:

In this paper, the authors present a novel computational method (DAESC) for performing differential allele specific expression analysis from single-cell RNA-seq data. The paper was a pleasure to read. The authors clearly articulate limitations of existing approaches and demonstrate how DAESC overcomes these limitation using both simulations and analysis of real data. I really appreciated the honest and straightforward reporting in the paper, for example when discussing the fact that type 1 error was not always strictly controlled at the desired nominal level. The code and data presented in the paper have been clearly documented and I did not detect any major methodological errors or omissions. I believe that DAESC is a useful tool that will be widely used by future single cell allele-specific expression studies.

I have only two extremely minor comments:

1. On lines 191-192 you state that "This indicates that cell-level variability, which is a special feature of single-cell ASE, could be important for implicit phasing". I think that this point is worth elaborating a bit more. Is it not possible to perform implicit phasing on bulk RNA-seq data, and if not, then why so?
2. On line 353 in the Discussion you mention the lme4 R package. I think it would be helpful to also add a citation to their preprint (<https://doi.org/10.48550/arXiv.1406.5823>) or the R package itself.

Response to Reviewer Comments

Reviewer #1 (Remarks to the Author):

This paper proposes a differential ASE analysis tool that considers two elements not examined by existing approaches and I consider it to be a very fascinating method. Despite the authors' mention of data size as a limitation, I believe that the current trend of scRNA-seq becoming more cost-effective will result in widespread adoption of this tool. The tool enables numerous analyses based on covariate variables such as cell type and trajectory, thereby enhancing its significance. The paper is well-written, and the data demonstrate the method's validity. Overall, the manuscript is well-written and can be published with minor revisions.

We thank the reviewer for the positive comments.

1) The perspective of the dispersion parameter was not discussed. In the case of single-cell RNA-seq, data quality issues such as zero-inflation can be considerable, and data quality is likely to influence the dispersion parameter. In general, sample size and variance parameter have an inverse relationship in terms of statistical power. I ask for comments on expected results regarding the data quality. For example, based on the simulation study, will glmm be superior when data quality is poor?

We conducted a new simulation study varying the overdispersion parameter ϕ and sequencing depth. We added the following text in Results (page 7 last paragraph):

“Next, we investigate the performance of DAESC under varying data quality, which is reflected by overdispersion parameter (ϕ) and sequencing depth. DAESC-BB and DAESC-Mix outperforms GLMM across varying levels of overdispersion (**Supplementary Figure 6**). Although all methods have lower power under strong overdispersion (large ϕ , low data quality), the advantage of DAESC of GLMM is also more pronounced (**Supplementary Figure 6**). We also observe that although DAESC-Mix is developed for large N (e.g., $N > 20$), it can also deliver strong performance under small N when the overdispersion is low (e.g., $N = 6$ and $\phi = 0.5$, **Supplementary Figure 6**), which is the case for many mouse datasets with low variance. In addition, we observe similar relative performance for DAESC-BB, DAESC-Mix and GLMM under 50%, 20%, and 10% sequencing depth of other scenarios, though all methods have lower power (**Supplementary Figure 7**).”

2) Experimental species determine the appropriate sample size in experimental design. In the analysis of mouse data with low variance, a sample size of 5 is already substantial. From this perspective, it would be beneficial to include species-specific guidelines or comments for the method.

In the new simulation study where we vary the overdispersion parameter (see response to comment 1), we also investigate the scenario of smaller sample size ($N = 6$). We observe that lower variance can compensate for small sample size. We added the following text in Results (page 8 paragraph 1):

“We also observe that though DAESC-Mix is developed for large N (e.g., $N > 20$), it can also deliver strong performance under small N when the overdispersion is low (e.g., $N = 6$ and $\phi = 0.5$, **Supplementary Figure 6**), which is the case for many mouse datasets with low variance.”

3) In Figure 1 and 2, A) -> a), which is not consistent with Figure 4. a)
We changed A) to a) in the legends of Figures 1 and 2.

4) In Figure 1, it would be better to relocate, either to top or bottom, the graph legend of c) and d).
We now relocated the legend to the bottom of Figure 1c.

5) In Figure 3-f, count -> Count in y-axis.
We changed “count” to “Count” in Figure 3f.

Reviewer #2 (Remarks to the Author):

The manuscript introduces a novel statistical method, DAESC, for differential allele-specific expression (ASE) analysis using single-cell RNA sequencing (scRNA-seq) data from multiple individuals. DAESC accounts for non-independence between cells from the same individual and incorporates implicit haplotype phasing. The method successfully identified dynamically regulated genes during endoderm differentiation and differentially regulated genes between type-2 diabetes patients and controls in pancreatic endocrine cells, providing new insights into gene regulation.

We thank the reviewer for their summary of our contributions.

Here are my comments:

1. It would be beneficial to provide a comprehensive tutorial outlining the complete analytical pipeline of scRNA-seq ASE analysis, from raw data to DAESC analysis. This should include steps such as GATK, pseudo-bulk, etc.

We have created a detailed step-by-step tutorial for the analytical pipeline: <https://github.com/qqi/DAESC/wiki>. We also included this link in the Code Availability section.

2. Although the authors demonstrated the robustness and superiority of DAESC over GLMM, the potential impact of preprocessing and upstream operations on their approach has not been evaluated. Factors such as different variant calling approaches, thresholds, scRNA-seq read depth, library size, 3' or 5', and platform differences should be systematically assessed and discussed. A systematic evaluation and an optimized, comprehensive analytical pipeline would be highly appreciated.

In general, we follow the preprocessing pipelines recommended by previous literature:

Castel, S. E., Levy-Moonshine, A., Mohammadi, P., Banks, E. & Lappalainen, T. Tools and best practices for data processing in allelic expression analysis. *Genome Biology* 16, 195 (2015).

Those include including alignment using STAR, variant calling and ASE read counting using GATK (<https://gatk.broadinstitute.org>), removing SNPs located in cross-mappable regions, and removing potential genotyping error. We evaluate the effects of A) read depth and B) removing genotyping error as follows.

A. Read depth.

We simulate ASE data with 50%, 20%, and 10% read depth of the other simulation studies. We describe the results as follows in Results – Simulation studies (page 8 paragraph 1):

“In addition, we observe similar relative performance for DAESC-BB, DAESC-Mix and GLMM under 50%, 20%, and 10% sequencing depth as other scenarios, though all methods have lower power (**Supplementary Figure 7**).”

B. Threshold for removing potential genotyping error

We evaluate the effect of this step in the endoderm differentiation data (Cuomo et al). We added the following text in Results - Dynamic ASE during endoderm differentiation (page 9 paragraph 2):

“We conduct two sensitivity analyses to evaluate the effects of analysis choices on the results. First, in the main analysis we remove SNPs with monoallelic expression to prevent false positives due to genotyping error. Here we repeat the same analysis with those SNPs included. We observe that removing SNPs with monoallelic expression ($\text{alt}/\text{total} < 0.02$ or $\text{alt}/\text{total} > 0.98$ in pseudobulk sample) have minimal effect on differential ASE except a small number of genes (<1% for both DAESC-BB and DAESC-Mix) that switches from significant to insignificant, or vice versa (**Supplementary Figure 12**).”

While we agree other factors such as variant calling approaches, 3' or 5', and platform differences may affect the results, their benchmarking is beyond the scope of this paper. Optimizing the preprocessing pipeline for ASE analysis overall is an important and complex topic that warrants a separate study. Our paper focuses on the statistical tests for post data processing. In addition, pipelines for obtaining even basic ASE estimates from 3' or 5' data are not well established. To avoid this issue, we choose to work with the full-length smart-seq2 data for our analysis.

Our sensitivity analyses show that read depth and batch effects typically do not change the relative performance between methods. We added the following text in Discussion (page 15 paragraph 1):

“Given the complexity of ASE data processing, other factors such as variant calling approaches, quality control thresholds, sequencing read depth and platform can also have effects on the results. A comprehensive evaluation of the factors and an optimized analysis pipeline is an important area of research. Though this is beyond the scope of the paper, we demonstrate through simulations that the relative ranks of the methods are robust to the change of overdispersion and read depth (**Supplementary Figures 6 and 7**), which are closely related to data quality.”

3. The simulation study comparing DAESC to GLMM seems biased to me, as it employs the same distribution (beta-binomial) as the authors' model. It would be valuable to investigate whether DAESC can outperform (or at least be comparable to) GLMM in simpler binomial cases.

We thank the reviewer for pointing this out. We have now conducted simulations using binomial GLMM (see Methods – Simulation studies for description of the setting), violating our models assumptions intentionally. We described the results as follows in Results – Simulation studies (page 8 paragraph 2):

“To evaluate the sensitivity of DAESC to model misspecification, we conduct another simulation study using binomial GLMM instead of beta-binomial (see Methods - Simulation studies). Theoretically, this scenario should give more advantage to the GLMM method. However, DAESC-BB and GLMM have nearly identical performance (**Supplementary Figure 8**). DAESC-Mix still leads to substantial power gain when there is low LD between the eQTL and the tSNP (**Supplementary Figure 8**).”

4. The comparison to only a GLMM model is limited, as other scRNA-seq and bulk data-based models have been developed. The authors should compare DAESC to a broader range of approaches in an unbiased manner.

We have now included other methods in the comparison. See Methods – Other methods for comparison (page 21-22) for description:

“We compare DAESC-BB and DAESC-Mix to other methods: GLMM, apeglm, apeglm-adj, EAGLE, and EAGLE-PB.”

“Apeglm is a fixed-effects beta-binomial regression:

$$y_{ij}|n_{ij} \sim BB(n_{ij}, \mu_{ij}, \phi), \quad \log\left(\frac{\mu_{ij}}{1 - \mu_{ij}}\right) = \beta_0 + \beta_1 x_{ij}$$

This model does not account for the sample repeat structure of single-cell ASE data. Therefore, we include a variation of apegglm (apeglm-adj) into the comparison, which further adjusts for donor IDs as fixed-effects covariates. Note that apegglm-adj can only be used for differential ASE with respect to a continuous variable but not binary case-control status, which is colinear with the one-hot encoding of donor IDs.

EAGLE¹ is another method developed for differential ASE analysis using bulk RNA-seq data. We first apply EAGLE directly to single-cell ASE data without accounting for the sample-repeat structure. For differential ASE across disease status, we further compare with EAGLE applied to pseudobulk data (EAGLE-PB). We aggregate cells from each individual into a pseudobulk sample by summing the alternative and total read counts. We then apply EAGLE to test for differential ASE using the pseudobulk samples.”

We describe the results in Results – Simulation studies (page 6 paragraph 2):

“In addition to GLMM, we compare DAESC with other methods, including beta-binomial regression implemented by apegglm¹ (also used in airpart¹), apegglm with donor IDs adjusted as covariates (apeglm-adj), EAGLE¹, and EAGLE applied to pseudobulk data (EAGLE-PB). See **Methods** for details. We observe inflated type I error for the raw apegglm and EAGLE due to failure to account for the sample repeat structure (**Supplementary Figure 3**). Apegglm-adj used fixed effects to account for sample repeat structure and have nearly identical performance as DAESC-BB for continuous cell states (**Supplementary Figure 3**). However, it cannot be applied to case-control comparisons since the case-control variable is colinear with the one-hot encoding of donor IDs. EAGLE-PB, the pseudobulk-based method for case-control comparisons, is less powerful than DAESC-BB especially when $r^2=0.1$ and 0.9 (**Supplementary Figure 3**). This shows the advantage of directly analyzing single-cell data over pseudobulk aggregation. EAGLE-PB assumes independent samples and is not applicable to the continuous-cell-state simulations shown in **Figure 1c** and **Supplementary Figure 3a**. The precision-recall curves show that DAESC-Mix dominates the other methods when $r^2=0$ and $N \geq 50$ with varying significance thresholds (**Supplementary Figure 4**), especially in the simulations for continuous cell states. In addition, the curves for the GLMM tend to dip near low recall value (**Supplementary Figure 4**), i.e., when the significant threshold is stringent. This indicates potential issues with p-value calibration. Nevertheless, GLMM appears to be the most comparable to DAESC-BB considering type I error and power, and its applicability to both continuous cell state and case-control comparisons. We use GLMM as the main comparison for the rest of the simulation studies.

5. Applying DAESC to cancer datasets, which typically exhibit more genomic changes and potential eQTLs, would demonstrate the versatility of the method and increase its relevance to the cancer research community.

Cancer cells have many somatic mutations which could violate the assumption of the model. For example, the implicit haplotype phasing provided by DAESC-Mix assumes all cells from a single individual have the same genotype. This assumption can be violated in cancer cells. Dealing with cancer datasets will require more sophisticated models which are beyond the scope of this paper. However, we discussed this issue in the Discussion section (page 15 paragraph 1):

“Lastly, DAESC is not specifically developed for analyzing cancer datasets. In particular, the implicit phasing in DAESC-Mix assumes every cell from an individual share the same genotype. This assumption is violated for cancer cells due to many somatic mutations. Single-cell ASE analysis in

cancer cells is also an intriguing future direction. ”

6. The authors model different cell types as independent variables. It would be interesting to explore whether a nested mixed model, with different cell types nested within individuals, could also be effective.

We thank the reviewer for this great suggestion. This is a fascinating direction that we would like to explore in the future. For this paper, however, we decided to restrict the scope to a simple comparison with respect to a pair of cells types, pseudotime, case-control status, etc. Due to the lack of statistical methods for single-cell differential ASE analysis, we believe this simple model still represents substantial progress from current literature. However, we agree with the reviewer a nested mixed model could be effective. We included the following comment in Discussion (page 15 paragraph 1):

“A future direction is to combine the strengths of DAESC and scDALI or airpart to incorporate sample repeat structure, implicit haplotype phasing and integration of information across cell types. A potential approach is to include cell types within individuals in a nested mixed model.”

7. Combining the DAESC-BB and DAESC-Mix models into a single function with an automatic switch or a hyperparameter for model selection could make the method more user-friendly and cohesive.

We thank the reviewer for this suggestion. We added a function `daesc` that conducts automatic selection between DAESC-BB and DAESC-Mix based on the number of donors. The GitHub README is updated with the following text:

“For automatic selection between `daesc_bb` and `daesc_mix`, use function `daesc`. `daesc` implements `daesc_bb` when the number of donors (N) is less than 20, and `daesc_mix` when $N \geq 20$.”

8. The haplotype phasing in the DAESC-Mix model using EM is a novel aspect of the paper. However, the accuracy of the EM estimation and the validation of the results should be addressed. Additionally, the ratio of $z=1$ and $z=-1$ estimated in real-world datasets should be discussed (considering 1 could be more common than -1).

A. Evaluation of phasing accuracy in simulation studies.

We conducted the same Fisher’s exact test of true vs. observed haplotype combinations for simulated data of 4,000 genes. True vs. observed phases are summarized by a contingency table like below for a gene, where each cell is the number of individuals in the category.

	$z_{true} = -1$	$z_{true} = 0$	$z_{true} = 1$
$z_{est} = -1$	9	18	1
$z_{est} = 1$	1	26	30

We conducted the test for genes with at least two true phases and two observe phases.

We included the follow text in Results – Simulation studies (page 6 paragraph 1):

“This is likely due to the ability of DAESC-Mix to conduct implicit haplotype phasing, which was shown to be effective overall (Fisher’s exact test p-value <0.05 in 36.5% genes tested, **Supplementary Figure 1**).”

B. Proportion of z=1 and z=-1 in real data

We added the following text in Results - Dynamic ASE during endoderm differentiation (page 9 last paragraph).

“For 48.4% of the genes that reach significance (FDR<0.05 by DAESC-Mix), DAESC-Mix learns two haplotype combinations with the minor haplotype including >10% individuals (**Supplementary Figure 13**).”

9. In many simulation studies, DAESC-BB shows similar results to GLMM in PR curves. This observation should be discussed and explained.

DAESC-BB is formulated as

$$\begin{aligned} y_{ij}|n_{ij} &\sim BB(n_{ij}, \mu_{ij}, \phi) \\ \log\left(\frac{\mu_{ij}}{1 - \mu_{ij}}\right) &= \beta_0 + \beta_1 x_{ij} + a_i \\ a_i &\sim N(0, \sigma_a^2) \end{aligned}$$

GLMM is formulated as:

$$\begin{aligned} y_{ij}|n_{ij} &\sim Binomial(n_{ij}, p_{ij}) \\ \log\left(\frac{p_{ij}}{1 - p_{ij}}\right) &= \beta_0 + \beta_1 x_{ij} + a_i + \epsilon_{ij} \\ a_i &\sim N(0, \sigma_a^2), \quad \epsilon_{ij} \sim N(0, \sigma_\epsilon^2) \end{aligned}$$

Both are binomial-based models that account for sample repeat structure and overdispersion. Their main difference is that the beta-binomial model accounts for overdispersion by incorporating the beta prior, while GLMM does so by a cell-specific random effect (ϵ_{ij}). Therefore, it can be expected that they would have similar PR curves in many scenarios.

We added the following text to Discussion (page 14 paragraph 2):

“In contrast, GLMM fitted by lme4 is more comparable to DAESC-BB than scDALI or airpart. Both GLMM and DAESC use random effects to model sample repeat structure but they account for overdispersion differently. Therefore, GLMM is used as the main reference method for benchmarking and has similar precision-recall curve to DAESC-BB in some of the scenarios (**Supplementary Figure 4**).”

However, in simulation studies, DAESC-BB outperforms GLMM when the data are generated from beta-binomial (**Figure 1**), and perform similarly to GLMM when the data are generated from GLMM (**Supplementary Figure 8**). Hence DAESC-BB has better overall performance.

10. The authors should conduct a larger number of simulations and report error bars (or confidence intervals) for the performance metrics.

We have now conducted 5,000 simulations for the main scenarios of one eQTL per tSNP. We added the following text in Methods (page 19 last paragraph):

“We repeat this procedure 10 times and obtain 5,000 simulations for each scenario (combination of N, r², differential ASE status). We observe minimal variation of type I error and power across 10 replications (**Supplementary Figure 15**). For the rest of the simulation studies, we conduct 400 simulations for each scenario to save computational time.”

11. The authors should investigate the computational time of DAESC across varying numbers of cells and patients in scRNA-seq data.

We conduct another simulation study to investigate the computational time of DAESC across datasets of varying sizes:

- Number of donors: 10,50,100,200.
- Average number of cells per donor: 100, 200, 300, 400.

We describe the results as follows in Results - Simulation studies (page 8 paragraph 2).

“We observe that though DAESC is computationally intensive due to its EM iterations, it can be easily handled by a modern computing cluster (see **Methods** for details and **Supplementary Figure 9** for results). For example, when analyzing a dataset of 200 individuals and on average 400 cells per individual (>2.5 times the size of the endoderm differentiation dataset in our application), DAESC-BB requires 3.3 hours to analyze 100 genes and DAESC-Mix requires 8.6 hours (**Supplementary Figure 9**).”

12. The potential impact of single-cell data quality, library size, and batch effects on the analysis should be discussed.

We thank the reviewer for raising this important point.

A. Batch effects

ASE analysis, which compares the expression of two alleles within a sample, typically protects against batch effects better than total gene expression. Our analysis further protects against batch effects by accounting for donor ID as random effects. To investigate remaining batch effects, we adjust for the month when the experiment was conducted. We described the results as follows Results - Dynamic ASE during endoderm differentiation (page 9 paragraph 2):

“Second, we evaluate whether additional batch effects may confound the analysis. After adjusting for the month when the experiment was conducted, the number of discoveries and the validation rate virtually remain the same (**Supplementary Table 3**).”

B. Data quality and library size

We added simulation studies to evaluate the effects. Please see our response to Comment 1 of Reviewer 1.

We added the following text in Discussion (page 15 paragraph 1):

“In addition, DAESC is focused on statistical analysis post data processing. Given the complexity of ASE data processing, other factors such as variant calling approaches, quality control thresholds, sequencing read depth and platform can also have effects on the results. A comprehensive evaluation of the factors and an optimized analysis pipeline is an important area of research. Though this is beyond the scope of the paper, we demonstrate through simulations that the relative ranks of the methods are robust to the change of overdispersion and read depth (**Supplementary Figures 6 and 7**), which are closely related to data quality.”

13. While the allelic imbalance findings across cell differentiation in Figure 4 are intriguing, the authors should provide related studies to validate their observations.

We conducted literature review and described the results in Results - Patterns and mechanisms of dynamic ASE (page 11 last paragraph):

“As a validation analysis, we examine whether our top 30 D-ASE genes (**Figure 4c**) have previously been reported to exhibit D-ASE, ASE, or other biological relevance in the literature. Moyerbrailean et al²³ found that 23 out of the 30 genes have ASE in cell types including lymphoblastoid cell lines (LCL), smooth muscle cells (SMC), murine erythroleukemia cells, HUVECs, and PBMCs. Fan et al¹⁴ reported 12 out of 30 genes have D-ASE in kidney, M0 macrophage cells, or M1 macrophage cells. Expression of some of the genes is tightly regulated in endodermal tissues. For example, *DKK1* was reported to be carefully regulated during kidney development²⁴; *GSTO1* was shown to have ASE in mouse lung, liver, and brain²⁵; and *GNAS* is a known imprinted gene in endodermal tissues such as pituitary²⁶, thyroid gland, and gonads²⁷.”

14. Validation strategies for the identified genes and their roles in cell differentiation and disease development should be discussed, as well as the reliability and robustness of the results.

We added a subsection in Methods (page 23):

“Validation of differential ASE genes

The list of dynamic eGenes reported by Cuomo et al¹⁰ can be used to validate our dynamic ASE findings. Since dynamic ASE is aimed to capture dynamically regulation of gene expression, dynamic ASE genes should have substantial overlap with dynamic eGenes. Therefore, we compare the proportion of significant dynamic ASE (FDR<0.05) that overlap with dynamic eGenes. To alleviate any doubt that different validation rates are caused by different number of genes identified by the methods, we create a concordance-on-top plot to compare the same number of top genes for all methods, which is varied from 10 to 800.”

The results are shown in Figure 3 and described in the Results (page 9 paragraph 1):

“Since D-ASE can be driven by dynamic cis-regulatory effects, we use the overlap between our D-ASE genes and dynamic eQTL genes reported by Cuomo et al¹⁰ as a validation criterion. Among the genes identified by DAESC-BB, 35.5% were reported by Cuomo et al, while among those identified by DAESC-Mix 27.5% were reported (**Figure 3d**). The GLMM identifies 19% fewer genes than DAESC-Mix (532 vs 657) and has a similar validation rate (**Figure 3d**). Comparing the same number of top genes (by smallest p-values) selected by each method, DAESC-Mix shows a higher validation rate than DAESC-BB or the GLMM across varying number of top genes (**Figure 3e**).”

We further validate the findings by literature review on previous findings on the genes - please see our answer to Comment 13. In addition, we validate the findings by gene-set enrichment analyses and found that the top 30 genes to be enriched in multiple pathways related to development and differentiation. See Figure 4 for details.

15. To further support the findings from real-world data, the authors should compare their results to those obtained using other approaches, such as scDALI and airpart. This will help alleviate any concerns regarding the validity of the results.

We ran scDALI with 10 expression PCs as cell states and donor IDs as fixed-effects covariates. We included a Venn diagram showing the number of D-ASE genes and overlap with DAESC-Mix (**Supplementary Figure 10**). We added the following text in Results - Dynamic ASE during endoderm differentiation (page 9 paragraph 1):

“scDALI finds 274 genes at $FDR < 0.05$, 77% of which are also found by DAESC-BB (Supplementary Figure 10).”

Reviewer #3 (Remarks to the Author):

In this paper, the authors present a novel computational method (DAESC) for performing differential allele specific expression analysis from single-cell RNA-seq data. The paper was a pleasure to read. The authors clearly articulate limitations of existing approaches and demonstrate how DAESC overcomes these limitation using both simulations and analysis of real data. I really appreciated the honest and straightforward reporting in the paper, for example when discussing the fact that type 1 error was not always strictly controlled at the desired nominal level. The code and data presented in the paper have been clearly documented and I did not detect any major methodological errors or omissions. I believe that DAESC is a useful tool that will be widely used by future single cell allele-specific expression studies.

We thank the reviewer for the positive comments.

I have only two extremely minor comments:

1. On lines 191-192 you state that "This indicates that cell-level variability, which is a special feature of single-cell ASE, could be important for implicit phasing". I think that this point is worth elaborating a bit more. Is it not possible to perform implicit phasing on bulk RNA-seq data, and if not, then why so?

We apologize for the confusion. We have now deleted this sentence.

2. On line 353 in the Discussion you mention the lme4 R package. I think it would be helpful to also add a citation to their preprint (<https://doi.org/10.48550/arXiv.1406.5823>) or the R package itself.

We added the reference to lme4 to the sentence (page 14 paragraph 2) in the Discussion: "In contrast, GLMM fitted by lme4³¹ is more comparable to DAESC-BB than scDALI or airpart."

Bates, D., Mächler, M., Bolker, B. & Walker, S. Fitting Linear Mixed-Effects Models using lme4. *arXiv.org* <https://arxiv.org/abs/1406.5823v1> (2014).

Additional Changes:

In the previous version, the GLMM used for analyzing endoderm differentiation data did not include cell-specific random effects for overdispersion. The GLMM for the simulation studies did include those random effects. To be consistent, we have now used the GLMM with those random effects (see Methods, page 21) to re-analyze the data and replaced the results in Figures 3 and 4. We also updated the description of these results in the text. The qualitative comparison remains consistent.

We moved the type I error and power figures for the case-control simulation setting to Figure 1, and moved the precision-recall curve for the continuous-cell-state simulations to Supplementary Figure 4.

Reviewers' Comments:

Reviewer #1:

Remarks to the Author:

I thank the authors for addressing my comments, and I have no further questions.

Reviewer #2:

Remarks to the Author:

I am pleased to note the thoroughness of the authors' revisions and the clarity in addressing my concerns. I have no more questions. Great work!